# Effect of Rearing Substrate on Nutritional Composition, Growth Performance and Multi-Omics Characteristics of Black Soldier Fly

**DOI:** 10.3390/insects17010010

**Published:** 2025-12-20

**Authors:** Kun Liu, Guangming Zhang, Yuting Li, Minghui Jiao, Jianlai Guo, Jun Li, Huibin Shi, Xianwei Wang, Weixian Zhang, Kai Quan, Wei Xia

**Affiliations:** 1Henan Key Laboratory of Unconventional Feed Resources Innovative Utilization, College of Animal Science and Technology, Henan University of Animal Husbandry and Economy, Zhengzhou 450046, China; liukun_139@163.com (K.L.); 13837147053@163.com (J.G.); lijun.nn@163.com (J.L.); huibinshi0715@163.com (H.S.); zhangwx126@126.com (W.Z.); quankai1115@163.com (K.Q.); 2College of Animal Science and Technology, Hebei Agricultural University, Baoding 071000, China; 15231488739@163.com; 3College of Animal Science and Technology, Henan Agricultural University, Zhengzhou 450046, China; liyuting08@163.com (Y.L.); m18239340797@163.com (M.J.); 4Henan Provincial General Station of Animal Husbandry Technology Extension, Zhengzhou 450000, China; wangxianwei2008@163.com

**Keywords:** black soldier fly, rearing substrate, protein feed, nutritional composition, growth performance, omics technologies

## Abstract

To address the global shortage of feed protein, this study tested how three rearing substrates—quail feed, food waste, and quail manure—affect the nutrition, growth, and related biological traits of black soldier fly (*Hermetia illucens* L.) larvae. The results showed food waste led to larvae with higher protein content, faster growth, and efficient material use; quail feed made larvae higher in fat; and the larvae reared on quail manure had the highest total mineral content. The rearing substrates also changed the larvae’s gut bacteria. We conclude that food waste is the most suitable tested substrate for large-scale larval production, as it balances nutrition and cost. This helps alleviate the current shortage of raw materials high in protein and offers a way to reuse food waste, benefiting both agriculture and the environment.

## 1. Introduction

In recent years, global stockbreeding has met with an urgent demand for dietary protein, and this trend is driven by multiple interrelated factors, including climate-induced crop yield reductions and shifts in global trade dynamics [1]. Therefore, identifying sustainable alternative protein sources has become a critical priority. Among various potential substitutes, insect-based feed has garnered significant international attention due to its unique advantages. The black soldier fly (BSF, *Hermetia illucens* L.), recognized as the most widely cultivated insect species globally, has garnered widespread international attention due to its rapid growth, low rearing cost, and efficient bioconversion of organic waste [2]. The BSF possesses several favorable traits, including a short life cycle, low rearing costs, and the ability to efficiently convert low-value organic waste—such as food scraps and livestock manure—into high-quality insect protein [3,4]. This “waste-to-value” capability not only mitigates environmental burdens but also opens new avenues for sustainable protein sourcing [5]. Research has demonstrated that incorporating BSFs into the diets of pigs [6], poultry [7,8,9] and aquaculture species [10,11] can fully or partially substitute for the nutritional role of fishmeal and soybean meal while enhancing animal growth performance.

The nutritional composition of BSFs is substantially influenced by the feeding substrate, with fat and protein contents being particularly sensitive to variations in the substrate matrix [12,13]. An optimal carbon-to-nitrogen (C/N) ratio in the substrate has been shown to not only enhance the crude protein content of BSF larvae but also promote efficient fat accumulation, thereby improving their potential as a natural antibacterial agent [14]. Nguyen et al. [15] evaluated six types of organic waste substrates—poultry feed, pig liver, pig manure, kitchen waste, fruit and vegetable waste, and fish-rendering waste—and found that insects reared on kitchen waste exhibited the highest average daily reduction rate as well as the greatest body length and weight (2.22 cm, 679.88 mg). Furthermore, larvae fed fish-rendering waste and pig liver demonstrated superior nutritional profiles compared to the control group, with a fat content reaching 11.6 g/100 g in the fish-rendering group and a histidine content totaling 21 g/100 g in the pig liver group. Additionally, the feeding substrate can modulate nutrient conversion efficiency, growth performance, and substrate conversion efficiency by shaping the intestinal microbiota of BSFs, thus influencing their suitability as an animal feed ingredient [16]. More specifically, three primary mechanisms underlie this modulation: First, functional microbial communities are selectively enriched based on the nutritional characteristics of the substrate. For example, high-fiber substrates such as rice straw promote the proliferation of fiber-degrading bacteria, including *Dysgonomonas* within the *Bacteroidetes* phylum [17,18]. Second, the microbial load of the substrate affects microbial homeostasis. Substrates with high microbial loads, such as chicken manure, may introduce exogenous pathogens and reduce the α-diversity of the gut microbiota [19], whereas low-load substrates, such as sterile feed, enable core bacterial populations to constitute over 70% of the community, resulting in a more stable and resilient structure [17]. Finally, the physicochemical properties of the substrate regulate the intestinal microenvironment. For instance, adjusting FW substrate particle size to achieve a pH of 5.71 ± 0.24 favors the dominance of *Lactobacillus* in BSF larvae, with a relative abundance of up to 95.2% [16].

Our previous research studied the application of live BSFs in quail production [20]. Then, in order to gain a more comprehensive understanding of the comprehensive application value of BSF in quail production, we conducted research on the utilization of BSFs in quail manure (QM). The high-protein dietary requirement (24–26% CP) of quails, combined with their relatively short digestive tract structure, led to a metabolic characteristic of “high intake-low retention”, resulting in a relatively high protein content in their manure. Diola et al. [21] pointed out that the crude protein content in quail manure can be as high as 32.60%, while, under the same experimental conditions, it is only 18.70% in chicken manure. Therefore, QM might have a better utilization efficiency and higher application value for BSFL rearing. However, despite numerous studies on feeding BSFL chicken manure, research on QM is scarce. Quail feed (QF) was chosen as a nutritionally consistent reference substrate (consideringthe variability of food waste and manure), and it represents a potential waste stream (e.g., spilled feed) on quail farms. Food waste (FW) was included as a commonly used reference substrate for BSFL. Therefore, we have selected these three substrates, which are easily accessible and commonly used in production. Previous studies have explored from multiple dimensions, but most of them focused on the functional exploration of specific substrates, the adaptation mechanisms of individual environmental stresses, or limited-dimensional nutritional analyses [22,23,24,25]. They have not yet conducted systematic comparisons among the three common and significantly different substrates: quail feed, food waste, and quail manure. Nor has the intrinsic regulatory network by which the substrate affects the nutritional composition and growth performance of larvae been deeply analyzed through multi-omics techniques. Therefore, this study evaluates the impact of three distinct substrates on the nutritional value and growth performance of BSFL, integrating analyses of intestinal microbiota, metabolites, and transcriptomic profiles, and explores the underlying mechanism of the use of substrate by BSFs. The findings may provide an empirical basis for future research and practical applications in BSF-based feed production.

## 2. Materials and Methods

### 2.1. Rearing and Sample Preparation

BSF eggs, laid on corrugated cardboard over less than 24 h, were obtained from Kuocheng Ecological Technology Co., Ltd. (Zhengzhou, China). The eggs were transferred to a substrate composed of bran and water at a 1:2 ratio and incubated in an artificial climate chamber under controlled conditions (30 °C, 65% relative humidity). Upon hatching, the BSFL were evenly distributed and reared on three distinct substrates. Each group consists of 3 replicates, with 1500 BSF larvae added to 1 kg of substrate for each replicate. The experimental treatments included quail feed (QF group), food waste (FW group), and quail manure (QM group), with three replicates per treatment. The QF substrate refers to the commercially purchased quail feed, while the FW substrate denotes the solid residue generated after impurity removal and three-phase separation (into oil, water, and solid phases) at the treatment facility, commonly known as kitchen and catering waste three-phase residue. To maintain optimal moisture levels for larval development, water was supplemented to the commercial feed substrate. In contrast, no additional water was added to the FW and QM substrates, as their natural moisture content was considered adequate to support larval growth [26]. The experiment was conducted in the Key Laboratory of Innovative Utilization of Unconventional Feed Resources in Henan Province, under controlled environmental conditions (28 °C ambient temperature, 70% relative humidity). Larval growth index (larval weight and length) was monitored daily, and freshly harvested larvae were collected and preserved for further analysis.

### 2.2. Nutritional Indicators

At the conclusion of the experiment, BSFL were separated from the frass using a sieve, the collected larvae were promptly and rapidly frozen with liquid nitrogen and stored at −20 °C pending chemical analyses. In this experiment, the nutritional indicators of feeding substrates, BSFL and frass were analyzed. Proximate analyses were performed according to Zulkifli et al. [27]. Crude protein content was quantified using the Kjeldahl method with an automated analyzer (Kjeltec 2300, FOSS, Hillerød, Denmark). Crude fat content was determined by gravimetric analysis following ether extraction using a Soxhlet apparatus (Soxtec 2043, FOSS, Hillerød, Denmark). Crude ash content was assessed by incinerating samples in a muffle furnace at 550 °C for 6 h and measuring the residual ash mass. Moisture content was calculated based on weight loss after drying samples at 105 °C to a constant weight.

Amino acid analysis was conducted according to Zulkifl et al. [27]. BSF samples were subjected to acid hydrolysis with 6N hydrochloric acid under anaerobic conditions. An aliquot of the hydrolysate was derivatized using AccQ-Fluor reagent with γ-aminobutyric acid (AABA) as the internal standard. The derivatized amino acids were analyzed using high-performance liquid chromatography (HPLC, Waters Alliance e2695, Waters Corporation, Milford, MA, USA) equipped with a fluorescence detector (Waters 2475) and AccQ-Tag C18 chromatographic column, and elute with a mobile phase containing AccQ-Tag eluent A (pH about 4.0) and acetonitrile according to a specific gradient program (0–10 min B phase 5–10%, 10–30 min 10–30%, 30–45 min 30–50%, 45–50 min 50–95%, 50–55 min 95–5%). Set the flow rate at 1.0 mL/min, column temperature at 37 °C, excitation wavelength at 250 nm, emission wavelength at 395 nm, and injection volume at 10 μL for detection. Amino acid concentrations were quantified using external calibration standards, with derivatization efficiency corrected based on the internal standard.

Fatty acid analysis was conducted in a manner similar to the research methodology outlined by Harwood [28]. Lipid hydrolysis was performed by refluxing samples with 1M potassium hydroxide in 95% ethanol, followed by extraction with a mixture of n-hexane and diethyl ether. The extract was centrifuged at 3000 rpm for 10 min, washed with distilled water, and the solvent was removed using a rotary evaporator to yield free fatty acids. Fatty acid methyl esters (FAMEs) were prepared by dissolving the extracted lipids in toluene and 1% sulfuric acid in methanol, followed by overnight incubation. After adding 5% sodium chloride solution, FAMEs were washed with 2% potassium carbonate solution, dried through anhydrous sodium sulfate, and diluted with n-hexane prior to gas chromatographic analysis. A DB-5 MS column (5% phenyl-95% dimethylpolysiloxane, weakly polar, with a regular specification of 30 m × 0.25 mm × 0.25 µm) was used. The chromatographic conditions were set as follows: high-purity helium gas (purity ≥ 99.999%) as the carrier gas, constant flow mode at 1 mL/min, injection port temperature at 280 °C, and splitless injection (1 μL injection volume, which can be adjusted to a split ratio of 10:1 according to the sample concentration). The temperature program was as follows: starting at 50 °C for 2 min, then increasing at 20 °C/min to 200 °C (no hold), then at 1 °C/min to 210 °C (no hold), and finally at 50 °C/min to 280 °C for 3 min. The mass spectrometry conditions were as follows: electron impact ionization source (EI), electron energy 70 eV, ion source temperature 200 °C, transfer line temperature 250 °C, solvent delay 4.5 min, and scan range m/z 50–500. This enabled the effective separation and detection of fatty acid methyl esters.

Mineral composition was determined using inductively coupled plasma optical emission spectrometry (ICP-OES, PerkinElmer Optima 5300 DV, PerkinElmer, Waltham, MA, USA) according to AOAC [29]. Sample digestion was carried out by weighing 0.1 g of dried BSF sample into a digestion tube, followed by the addition of 5 mL of concentrated nitric acid (69%). The mixture was heated in a fume hood until white fumes appeared. After cooling, 3 mL of hydrochloric acid was added, and the solution was further heated until the color changed from green to yellow/orange/red, followed by an additional 10-minute heating to ensure complete digestion. The digested solution was cooled, transferred to a 100 mL volumetric flask, and diluted to volume with deionized water. The solution was allowed to settle before instrumental analysis.

### 2.3. Growth Indicators

Twenty BSFL exhibiting consistent growth and developmental stages were randomly selected daily from each replicate group to determine average body weight [26]. Additionally, ten larvae of uniform size were selected from each group to measure average body length. At the conclusion of the experiment, samples of larval frass, substrate, and dried larvae were collected and oven-dried at 105 °C until reaching a constant weight, in order to calculate moisture content for each sample. Based on these measurements, the following parameters were calculated according to predefined formulas:$$LY(\%)=\frac{{LW}_{W}}{\;{SW}_{W}\;}\times 100$$$${SR}_{W}\;\left(\%\right)=\frac{\left(SW-FW\right)}{SW}\times 100$$$${SC}_{W}\;(\%)=\frac{BW}{\left(SW-FW\right)}\times 100$$$${SR}_{D}\;\left(\%\right)=\frac{\left(SDM-FDM\right)}{SDM}\times 100$$$${SC}_{D}\;(\%)=\frac{BDM}{\left(SDM-FDM\right)}\times 100$$
where LY denotes larvae yield, LW_W_ denotes wet weight of larvae, SW_w_ denotes wet weight of substrate, SR_W_ denotes substrate fresh weight reduction, SW denotes initial substrate fresh weight, FW denotes frass fresh weight, SC_W_ denotes Substrate fresh weight conversion, BW denotes insect body fresh weight, SR_D_ denotes substrate DM reduction, SDM denotes initial substrate DM, FDM denotes frass DM, SC_D_ denotes Substrate DM conversion, and BDM denotes insect body DM.

### 2.4. Microbiota Collection, 16S rRNA Sequencing, and Bioinformatic Analysis

Following extraction of BSF larval intestinal tracts under laminar flow, specimens were cryopreserved at −80 °C. DNA was isolated using FastDNA^®^ Spin Kit (MP Biomedicals, Santa Ana, CA, USA) per manufacturer guidelines, with integrity verified by agarose electrophoresis and concentration quantified via NanoDrop 2000 (Thermo Fisher Scientific, Wilmington, DE, USA). The 16S rRNA V3-V4 regions were amplified in triplicate 20 μL PCR mixtures (4 μL 5× TransStart buffer, 2 μL dNTPs, 0.8 μL primers 338F/806R, 0.4 μL polymerase, 10 ng DNA) using ABI 9700 thermocycler (Applied Biosystems, Waltham, MA, USA): 95 °C/3 min; 27 cycles (95 °C/30 s, 55 °C/30 s, 72 °C/45 s); 72 °C/10 min. Amplicons were purified (NEXTflexTM kit, Bioo Scientific, Austin, TX, USA), quantified (Quantus™ Fluorometer, Promega, Madison, WI, USA), and sequenced on Illumina MiSeq PE300 (Illumina, San Diego, CA, USA) at Shanghai Majorbio Bio-pharm Technology Co., Ltd. (Shanghai, China). Raw data underwent quality control with fastp v0.20.0 [30], read merging using FLASH v1.2.7 [31], and OTU clustering at 97% similarity via UPARSE v7.1 with chimera removal [32]. Taxonomic assignment was performed with RDP Classifier v2.2 against SILVA v138 (0.7 confidence threshold) [33].

### 2.5. Untargeted Metabolic Profiling by UPLC-QTOF/ESI-MS

The collected insects were disinfected with sterile water and cryopreserved at −80 °C. For metabolite extraction, 50 mg quail body samples were homogenized in 2 mL tubes with 6 mm grinding beads, using 400 μL ice-cold methanol/water (4:1, *v*/*v*) containing 0.02 mg/mL L-2-chlorophenylalanine as internal standard. Mechanical disruption was performed in a Wonbio-96c grinder (Wonbio Technology, Nanjing, Jiangsu, China) (−10 °C, 50 Hz, 6 min) followed by ultrasonic extraction (5 °C, 40 kHz, 30 min). After 30-minute incubation at −20 °C, samples were centrifuged (13,000× *g*, 4 °C, 15 min), and supernatants were analyzed by UHPLC-Q Exactive HF-X system (Thermo Scientific, Waltham, MA, USA) with ACQUITY HSS T3 column (100 mm × 2.1 mm i.d., 1.8 µm; Waters, Milford, MA, USA) at Majorbio Bio-Pharm Technology Co. Ltd. (Shanghai, China). Mobile phase consisted of 0.1% formic acid in water/acetonitrile (95:5, *v*/*v*; A) and acetonitrile/isopropanol (47.5:47.5, *v*/*v*; B) with 0.1% formic acid, flowing at 0.40 mL/min (40 °C). The injection volume was 3 μL. MS detection employed dual ESI (±3500 V) with source temperature at 425 °C; sheath gas flow rate at 50 arb; Aux gas flow rate at 13 arb; ion-spray voltage floating (ISVF) at −3500 V in negative mode and 3500 V in positive mode, respectively; Normalized collision energy, 20-40-60 V rolling for MS/MS. Full MS resolution was 60,000, and MS/MS resolution was 7500. Data acquisition was performed with the Data Dependent Acquisition (DDA) mode. The detection was carried out over a mass range of 70–1050 m/z. Progenesis QI processed raw data after removing internal standards and QC-failed features (RSD > 30%), with metabolites annotated via HMDB, Metlin, and Majorbio databases. Multivariate analysis using ropls (v1.6.2) identified significant metabolites (VIP > 1, *p* < 0.05) through PCA and OPLS-DA (7-fold CV), followed by KEGG pathway analysis with scipy.stats (v1.16.2).

### 2.6. Transcriptome Analysis

Total RNA was isolated from BSFL tissues using TRIzol^®^ Reagent (Thermo Fisher Scientific, Waltham, MA, USA) following manufacturer protocols, with RNA integrity verified by 5300 Bioanalyser (Agilent Technologies, Santa Clara, CA, USA) and concentration quantified via ND-2000 (NanoDrop Technologies, Wilmington, DE, USA). Only specimens meeting strict quality criteria (OD260/280 = 1.8–2.2, OD260/230 ≥ 2.0, RQN ≥ 6.5, 28S:18S ≥ 1.0, >1 μg) proceeded to library construction. Sequencing libraries were prepared using Illumina^®^ Stranded mRNA Prep (Illumina, San Diego, CA, USA) with 1 μg total RNA through polyA selection, fragmentation, cDNA synthesis (random hexamer priming), end-repair, adapter ligation, and 300–400 bp insert size selection (15 PCR cycles). Libraries were sequenced on NovaSeq X Plus (PE150, Illumina, San Diego, CA, USA) following Qubit 4.0 quantification. Raw reads underwent quality control via fastp (v0.23.2) [30], followed by HISAT2 (v2.2.1) [34] genome alignment and StringTie (v2.2.1) [35] assembly. Differential expression analysis employed RSEM (v1.3.3) [36] quantification with DESeq2 (v1.38.3) [37] or DEGseq (v1.42.0) [38] for statistical evaluation (|log2FC| ≥ 1, FDR < 0.05/0.001). Functional enrichment (GO/KEGG) used Goatools (v1.2.5) and scipy (v1.16.2) (*p* < 0.05 Bonferroni-corrected), while alternative splicing events were detected by Rmats (v4.1.2) [39] considering exon inclusion/exclusion and intron retention.

### 2.7. Statistical Analysis

The experimental data were organized using Excel 2013 software, and the significance of differences among data was analyzed with GraphPad Prism 8. The results are presented as “mean ± standard deviation,” with *p* values indicated for statistical significance (*p* < 0.05). No letters denote groups that did not show significant differences. In the bar chart, * indicates *p* < 0.05 and ** indicates *p*< 0.01.

## 3. Results

### 3.1. Nutritional Components

The fundamental nutritional components of the rearing substrate are presented in Table 1. In terms of total ash content, QM had the highest content (24.6 ± 0.1%), significantly higher than QF (12.6 ± 0.1%) and FW (11.3 ± 0.1%). The crude protein content showed a significant gradient of QM (41.3 ± 0.4%) > FW (35.2 ± 0.7%) > QF (21.2 ± 0.2%). The crude fat content of QF (10.6 ± 0.3%) and FW (6.9 ± 0.1%) was similar without significant difference, and both were significantly higher than that of QM (2.7 ± 0.1%), which belongs to a low-fat matrix. The calcium content was the highest in QM (5.1 ± 0.1%), followed by QF (3.6 ± 0.1%), and the lowest in FW (2.1 ± 0.1%). In terms of phosphorus content, QM (2.2 ± 0.2%) was significantly higher than QF and FW (both 0.6 ± 0.1%), while there was no significant difference between the latter two. As presented in Table 2, the basic nutritional profiles of BSFL were significantly influenced by the type of rearing substrate. Moisture content varied significantly among groups, with the highest value observed in the FW group, which was significantly higher than that in the QF and QM groups (*p* < 0.05). The QM group exhibited the highest ash content, significantly exceeding that of both the QF and FW groups (*p* < 0.05). No significant difference was found in crude protein content between the FW and QM groups, both of which were significantly higher than that of the QF group (*p* < 0.05). Crude fat content was highest in the QF group (33.2 ± 1.3%), followed by the FW group (25.1 ± 1.1%), and lowest in the QM group (7.0 ± 1.1%), with statistically significant differences among all three groups (*p* < 0.05). Calcium content was highest in the FW group, while phosphorus content was lowest in this group. A significant difference in calcium content was observed between the QF and QM groups, although no such difference was found for phosphorus content. Regarding larval frass composition (Table 3), the FW group showed the highest moisture content (85.1 ± 0.1%). Crude fat and crude protein levels were highest in the QF and FW groups, with the QM group showing significantly lower values. In contrast, calcium content in the frass was highest in the QM group, while phosphorus content peaked in the QF group. Both calcium and phosphorus levels were lowest in the FW group, with significant differences observed among all groups.

The results of amino acid composition analysis of BSFL raised on different substrates (Table 4) showed that there was no significant difference in the total essential amino acid content among the three samples, and the contents of most essential amino acids such as isoleucine, leucine and lysine also showed no significant difference. However, there were significant differences in the contents of valine, phenylalanine and threonine (*p* ≤ 0.05). The contents of valine and threonine in FW and QM were significantly higher than that in QF, and the content of phenylalanine in FW was significantly higher than that in QF and QM. There was a significant difference in the total non-essential amino acid content. The contents of FW and QM (24.0 ± 0.3% and 24.4 ± 0.7% respectively) were significantly higher than that of QF (21.4 ± 1.1%). Specifically, the contents of aspartic acid, serine, glycine and arginine in FW and QM were significantly higher than that in QF, the content of histidine in FW and QM was significantly higher than that in QF, the content of tyrosine in FW was significantly higher than that in QM, and the content of glutamic acid in QM was significantly higher than that in FW and QF. However, there was no significant difference in the contents of proline and alanine among the three samples. The fatty acid analysis of BSFL (Table 5) revealed that the major fatty acid components were lauric acid (C12:0), palmitic acid (C16:0), linoleic acid (C18:2n6t), and arachidic acid (C20:0). The FW group exhibited the highest total saturated fatty acid content, which was significantly higher than that of the QM and QF groups. The QF group showed the highest levels of C10:0, C12:0, and C14:0, with statistically significant differences compared to the QM group (*p* < 0.05). The QM group had the highest content of C16:0, C17:0, and C20:0, significantly greater than that in the other two groups. The FW group had the highest C21:0 content (4.3 ± 0.6%), which was significantly higher than that in both the QF and QM groups. Regarding unsaturated fatty acids, the QF group exhibited the highest total unsaturated fatty acid content (21.5 ± 0.9%), with C18:2n6t levels (17.7 ± 0.7%) significantly higher than those in the FW group (0.3 ± 0.1%) and QM group (0.7 ± 0.1%) (*p* < 0.05). The QM group displayed the highest contents of C14:1, C16:1n7, and C18:1n9t among all groups, with statistically significant differences compared to the QF and FW groups (*p* < 0.05). As presented in Table 6, the QM group exhibited the highest content of all mineral elements, with highly significant differences compared to both the QF and FW groups (*p* < 0.05). The FW group showed significantly higher levels of iron (Fe) and sodium (Na) compared to the QF group, whereas the QF group had significantly higher concentrations of zinc (Zn), manganese (Mn), and magnesium (Mg) than the FW group (*p* < 0.05).

### 3.2. Growth Indicators

As illustrated in Figure 1A, the BSFL in the FW group exhibited the fastest growth rate in body length and achieved the greatest length on the seventh day of collection, which was significantly higher than that of the other two groups (Figure 1B). The body length of the QF group was significantly greater than that of the QM group on days 1, 4, and 5; however, no significant differences were observed after day 6 (*p* > 0.05). The body weight dynamics of BSFL (Figure 1C) were generally consistent with the trends observed in body length. On the seventh day of collection, the FW group had the highest body weight, followed by the QF group and then the QM group, with significant differences among all groups (*p* < 0.05) (Figure 1D). On day 2, the QF group showed significantly higher body weight compared to the other two groups. From day 3 onward, the FW group demonstrated the fastest weight gain, which was significantly higher than that of the QF and QM groups (*p* < 0.05). Between days 3 and 6, no significant differences in body weight were observed between the FW and QM groups. However, by day 6, the QF group exhibited significantly higher body weight than the QM group. Figure 1E presents the effects of three different substrates on the production performance of BSFL. Both the QF and FW groups showed significantly higher fresh insect production compared to the QM group. The QF group exhibited the highest reduction rate, significantly exceeding the other two groups. Furthermore, the FW group demonstrated significantly higher fresh weight conversion rate, dry matter reduction rate, and overall conversion efficiency compared to both the QF and QM groups (*p* < 0.05).

### 3.3. Microbiome

Analysis of the intestinal microbiota of BSFL reared on three different substrates revealed distinct compositional differences among the three groups, with the QM group exhibiting the most pronounced inter-group variation (Figure 2A). Venn diagram analysis at the genus level (Figure 2B) indicated that the QF and FW groups had 6.37% and 6.13% unique microbial genera, respectively, whereas the QM group had a significantly higher proportion of unique genera (57.31%). All three groups shared the genera *Enterococcus*, *Actinomyces*, and norank_f_norank_o_RsaHf231. Compared to the FW and QM groups, the QF group showed higher abundances of *Escherichia-Shigella*, *Lactobacillus*, and *Pediococcus*. The FW group was characterized by the presence of *Salana* and *Dysgonomonas*, while the QM group exhibited increased levels of *Achromobacter* and *Pseudomonas* (Figure 2C). At the genus level, *Enterococcus* was the most dominant, accounting for 51.91% across all groups, followed by *Actinomyces* at 9.09% (Figure 2D). Heatmap analysis (Figure 2E) demonstrated that *Escherichia-Shigella*, *Lactobacillus,* and *Pediococcus* were significantly enriched in the QF group, whereas *Salana* and *Dysgonomonas* were significantly higher in the FW group. *Actinomyces* showed the highest abundance in the QM group, significantly exceeding that in the other two groups. Interaction network analysis (Figure 2F) revealed a positive correlation between *Enterococcus* and *Lactobacillus*, *Pediococcus*, and *Escherichia-Shigella*. *Actinomyces* was positively correlated with norank_f_norank_o_RsaHf231 and *Dysgonomonas*. *Dysgonomonas* showed positive correlations with *Actinomyces* and *Salana*, among others, across six genera.

### 3.4. Metabolome

Metabolomic analysis of BSFL revealed that different rearing substrates had a significant impact on the composition of metabolites, leading to the formation of distinct metabolic profiles among the groups (Figure 3A). Notably, the QM group exhibited a significantly higher diversity of metabolites compared to the other groups (Figure 3B). Differential metabolite analysis among the three groups (Figure 3C) identified a total of 29 significantly enriched KEGG pathways. Between the QF and QM groups, 573 metabolites were upregulated and 693 were downregulated, with significant enrichment observed in the pathways of chemical carcinogenesis-DNA adducts, Retrograde endocannabinoid signaling, and Bile secretion (Figure 3D). Comparing the QF and FW groups, 296 metabolites were upregulated and 321 were downregulated, showing significant enrichment in tryptophan metabolism and bile secretion pathways (Figure 3E). Between the QM and FW groups, 565 metabolites were upregulated and 422 were downregulated, with all of the top 20 enriched pathways showing statistically significant enrichment (*p* < 0.05) (Figure 3F).

### 3.5. Transcriptome

Transcriptomic analysis of BSFL, based on PCA results, demonstrated that the QF and FW groups exhibited highly similar gene expression profiles, whereas the QM group showed distinct separation from the other two groups (Figure 4A). Venn diagram analysis (Figure 4B) revealed that 69 genes were uniquely expressed in the QF group, 372 in the QM group, and 621 in the FW group. Differential gene expression analysis among the three groups (Figure 4C) indicated that the QF vs. QM comparison showed the most significant differences, with 1285 upregulated and 571 downregulated genes. These differentially expressed genes were enriched in key pathways such as toll and imd signaling pathway, Estrogen signaling pathway, and Galactose metabolism (Figure 4D). In the QF vs. FW comparison, 229 genes were upregulated and 208 were downregulated, which were associated with pathways including glycolysis/ gluconeogenesis, pyruvate metabolism, and pentose phosphate pathway (Figure 4E). For the FW vs. QM comparison, 101 genes were upregulated and 230 were downregulated, and these genes were enriched in functions related to DNA replication, toll and imd signaling pathway, and tyrosine metabolism (Figure 4F).

## 4. Discussion

The composition and content of crude protein, crude fat, and mineral elements in BSFL are primarily influenced by the type and quantity of its rearing substrate [40]. Studies have shown that higher substrate protein content correlates with increased larval protein levels, while fat content is mainly influenced by the carbohydrate content of the substrate [41]. In this study, the crude protein content of BSF in the FW and QM groups was significantly higher than that in the QF group (47.5–48.3% vs. 37.4%), whereas the QF group exhibited the highest crude fat content (33.2%). These findings align with previous reports that larvae reared on high-protein substrates (e.g., vegetables, brewery by-products) show elevated crude protein levels, while those fed high-carbohydrate substrates (e.g., starch, quail feed) accumulate more fat [42,43]. On the other hand, larvae reared on QM had high contents of protein and ash, with low lipids, characteristic of larvae reared on manure-based substrates, i.e., low carbohydrate and high mineral content [13]. The subsequent multi-omics analyses will therefore build upon these well-defined nutritional phenotypes to uncover the underlying molecular mechanisms driving these adaptations.

The larvae reared on FW and QM had higher contents of several essential amino acids, with the FW group exhibiting higher levels of lysine and threonine. Notably, lysine and threonine—both key limiting amino acids in animal feed—were present in higher amounts in FW-reared larvae. Sandrock et al. [44] demonstrated that the levels of lysine and threonine in BSF were significantly affected by the substrate. The abundance of these amino acids directly enhances the nutritional value of BSFL meal, potentially reducing the need for synthetic amino acid supplementation in animal diets [6]. This balanced amino acid composition indicates that FW-derived protein may serve as a more nutritionally complete source. Conversely, the relative deficiency of certain essential amino acids in the QF group may reflect an imbalanced substrate or a metabolic shift towards fat synthesis at the expense of optimal protein deposition.

The fatty acid composition and lipid content of BSFL are also influenced by the carbohydrate content in the substrate [45]. BSFL fat is predominantly composed of saturated fatty acids, particularly C12:0, C14:0, C16:0, and C18:0 [12,46], In terms of fatty acid composition, lauric acid (C12:0) was the predominant saturated fatty acid, particularly in the QF and FW groups. Lauric acid is renowned for its antimicrobial effects on the gut microbiota, particularly its activity against Gram-positive bacteria [47]. Furthermore, it exhibits inhibitory activity against viruses, bacteria, and parasites, demonstrating the potential to improve gut health and enhance the immune status of animals [48]. The QF group also exhibited a higher proportion of unsaturated fatty acids, such as linoleic acid (C18:2n6t), which may offer nutritional benefits in feed formulations. The distinct fatty acid profiles across substrates highlight the remarkable plasticity of BSFL lipid metabolism. This adaptability not only allows for the targeted enrichment of high-value functional fatty acids but also underscores the potential of BSFL as a sustainable source for biodiesel production, given the superior oxidative stability of its lipid profile [49,50].

Mineral analysis indicated that the QM group accumulated significantly higher levels of trace elements such as Fe, Zn, and Mn, consistent with studies using poultry manure [51]. While this enhances the mineral profile of larvae, it also necessitates careful consideration of inclusion rates in feed to avoid mineral toxicity. The FW group showed elevated Fe and Na levels, likely reflecting the salt and hemoprotein content typical of food waste.

Insect frass, a mixture of undigested substrate, excrement, and larval debris, can be further processed through anaerobic digestion, pyrolysis for biochar production, or traditional composting, thereby enhancing the economic value of insect farming on a larger scale [52,53]. Zhu et al. [54] investigated a similar sequential process involving fungal treatment of agricultural waste followed by composting to improve resource utilization efficiency. Direct application of insect frass as a substitute for mineral fertilizers or vermicompost has also been proposed as a viable strategy [55,56]. In this study, the QF group exhibited the highest nutrient and mineral content in frass, with phosphorus-essential for plant growth-being most abundant in this group, making it the most suitable for agricultural use as organic fertilizer.

In terms of growth performance, both body length and weight were highest in the FW group. Bellezza et al. [57] reported that within a lipid content range of 1% to 4.5%, higher lipid levels correlated with faster BSFL growth, with developmental differences primarily attributed to lipid availability. In addition to absolute lipid content, the lipid-to-protein ratio significantly influences growth performance. In this study, the FW group demonstrated superior growth performance, including the highest body length, weight, and dry matter conversion efficiency. This aligns with findings that easily degradable substrates rich in carbohydrates support rapid larval development and efficient biomass conversion [42,43]. The lower performance in the QM group may be attributed to the substrate’s complex structure and lower digestibility, highlighting the importance of substrate physical and chemical properties in larval development.

The microbial community structure of BSFL is shaped by environmental nutrient composition and the microbial profile of the rearing substrate, with microorganisms entering the gut primarily through ingestion [17]. The gut microbiota of BSFL varied significantly across substrate groups, exhibiting both a stable core and substantial plasticity in response to diet. We identified several previously reported genera, including *Dysgonomonas*, *Morganella*, *Enterococcus*, *Pseudomonas,* and *Actinomyces*, which are considered core components of the BSF intestinal microbiota [17]. In their study, *Dysgonomonas*, *Actinomyces*, and *Enterococcus* accounted for 44% of total reads and were present in over 80% of samples. In our study, *Enterococcus* constituted 51.91% of the reads, and *Actinomyces* accounted for 9.09%. *Dysgonomonas*, typically among the top three most abundant genera [58,59], was only present at 4.91% and absent in the QF group. These findings suggest that *Enterococcus* and *Actinomyces* are highly conserved in BSFL, while *Dysgonomonas* is more susceptible to substrate-driven shifts. Despite its variability, *Dysgonomonas* plays a crucial role in the insect gut [60,61,62]. As a Gram-positive facultative anaerobe, *Enterococcus* facilitates nutrient absorption and supports intestinal homeostasis [63]. *Actinomyces*, known for degrading complex organic compounds including lignin and chitin [64], also produces antimicrobial substances that suppress pathogenic bacteria, benefiting larval health [65,66]. *Dysgonomonas* is well known for its role in lignocellulose degradation in termite guts and is also prevalent in BSFL intestines [67,68]. Metatranscriptomic analysis by Kariuki et al. [23] revealed that *Dysgonomonas* and *Bacteroides* were most abundant in BSFL fed lignocellulose-rich substrates. Bruno et al. [69] highlighted its importance in complex polysaccharide digestion, and Jiang et al. [18] further demonstrated its positive correlation with genes involved in sulfate, carbohydrate, and nitrogen metabolism. Shelomi et al. [70] identified *Dysgonomonas* as a core gut microbiota member, present across diverse waste substrates. Metagenomic studies have also revealed novel non-lactonase genes in *Dysgonomonas* strains capable of breaking down indigestible plant carbohydrates [71]. Therefore, *Dysgonomonas* abundance correlates positively with waste and lignocellulose content. *Enterococcus* and *Actinomyces* were identified as core taxa of the gut microbiota of BSFL. The high abundance of *Actinomyces* in the QM group may reflect its role in degrading complex organic compounds and suppressing pathogens, thus maintaining gut homeostasis under high microbial load [63,64,65]. The presence of *Dysgonomonas* in the FW and QM groups—but not in QF—correlates with lignocellulose content, supporting its established role in polysaccharide degradation and nutrient metabolism [66,67,68]. The absence of *Dysgonomonas* in the QF group may explain the lower efficiency in fiber-rich substrate utilization.

Differential metabolites across groups were mapped to 29 KEGG pathways. Metabolomic profiling revealed that the QM group exerted the most pronounced effect on larval metabolism, with significant enrichment observed in pathways including bile secretion and retrograde endocannabinoid signaling. The enrichment of the retrograde endocannabinoid signaling pathway in the QM group may reflect a physiological response to environmental stress or pathogenic challenge, which is consistent with the high microbial load characteristic of manure [72]. In contrast, the FW and QF groups exhibited enrichment in tryptophan metabolism and carbohydrate-related metabolic pathways, respectively. Tryptophan, an essential amino acid, is mostly used by the body to synthesize a variety of biologically active compounds [73]. These compounds play a crucial role in regulating the host’s immune response, neural function, intestinal homeostasis and inflammatory levels [72]. Therefore, the alteration of the tryptophan metabolic pathway may indicate that the larvae are actively mobilizing their physiological defense and homeostasis maintenance systems. Jing et al. [74] demonstrated that *Enterococcus* contributes to amino acid biosynthesis, protein digestion, energy metabolism, and detoxification, supporting the observed metabolic differences. These findings suggest that the increased abundance of *Enterococcus* in the larval gut microbiota promotes the enrichment of tryptophan metabolism and carbohydrate-associated pathways. In contrast, the QM group showed enrichment in disease-related pathways such as retrograde endocannabinoid signaling, suggesting increased pathogenic potential. Transcriptomic analysis further supported these findings, with the QM group exhibiting upregulation of immune-related pathways (e.g., Toll and Imd signaling), likely as a defense mechanism against manure-borne pathogens. This indicates that larvae in the QM group invest significant energy in immune defense, which may divert resources away from growth and biomass accumulation [58]. In contrast, the FW and QF groups showed enrichments in metabolic pathways related to energy and nutrient assimilation, such as glycolysis/gluconeogenesis and pyruvate metabolism, aligning with their superior growth and conversion efficiencies. The upregulation of genes related to DNA replication and tyrosine metabolism in the FW group further supports its rapid growth rates and cuticle sclerotization processes [75]. The above joint analysis of BSF microorganisms, metabolites and transcriptomes is conducive to a better understanding of the molecular mechanism by which BSF utilizes substrates.

From a practical standpoint, FW emerges as the most suitable among the three substrates for BSFL production, providing an optimal balance among nutritional quality, growth performance. Although QM enhances mineral content, its application requires careful management to mitigate potential biosafety risks. QF supports high levels of larval lipids and unsaturated fatty acids; however, it may be less economically viable for industrial-scale implementation. Future studies should focus on validating these findings in field conditions, evaluating the effects of BSFL-based feeds on livestock health and productivity, and exploring the potential of frass as an organic fertilizer to enhance the circular economy in agricultural systems.

## 5. Conclusions

This study demonstrates that food waste offers the best balance in the nutritional composition and growth performance of BSF among the tested substrates, underpinned by a favorable gut microbiome (e.g., enriched *Dysgonomonas*), efficient metabolic pathways for nutrient assimilation, and transcriptomic profiles supporting rapid growth. Quail manure promotes mineral accumulation in the larvae, as reflected in its distinct metabolomic profile and transcriptomic upregulation of immune-related pathways, indicating an adaptive response to the high-microbial-load environment. Quail feed is advantageous in terms of specific fatty acid profiles, though its associated microbiota and metabolic pathways showed lower efficiency in converting complex substrates. The multi-omics data collectively reveal that the rearing substrate shapes the larval phenotype not only through direct nutrient provision but also by remodeling the gut microbiome and subsequent host metabolic and transcriptional responses. These findings provide an empirical basis for the selection of BSF rearing substrates.

## Figures and Tables

**Figure 1 insects-17-00010-f001:**
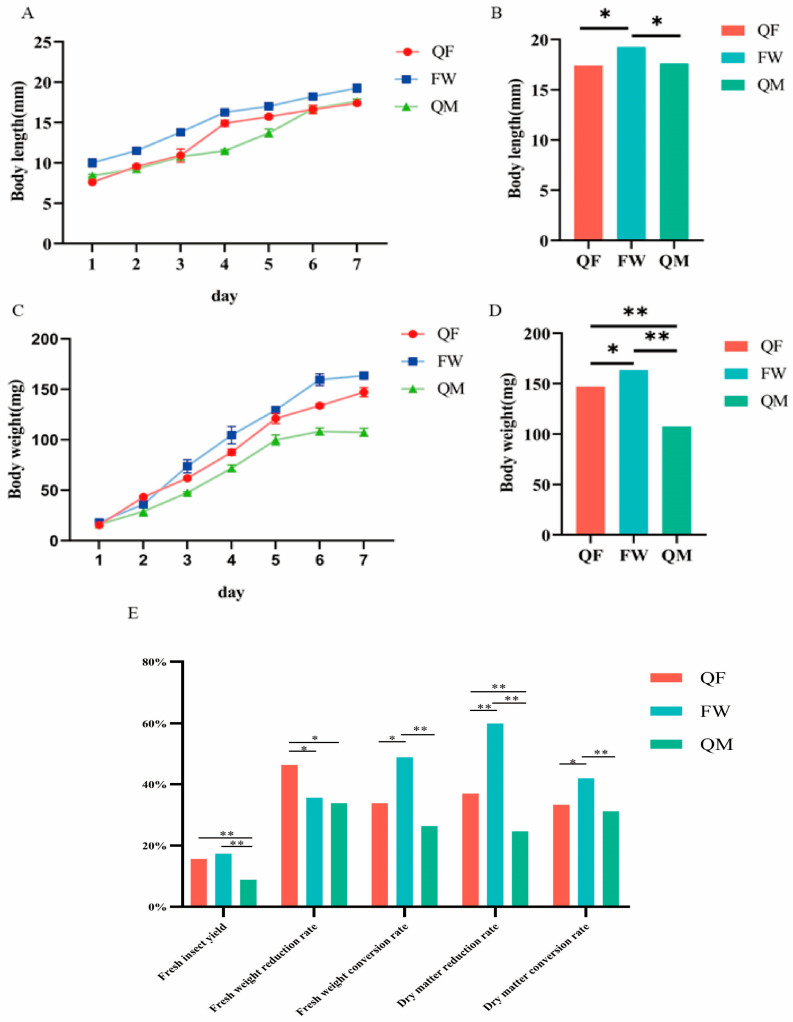
Effects of three different rearing substrates on the body length, body weight and production performance of BSFL. (**A**) The length of the insect body from day 1 to day 7. (**B**) The body length of BSFL on day 7 of collection. (**C**) The weight of the insect body from day 1 to day 7. (**D**) The body weight of BSFL on day 7 of collection. (**E**) Production performance. *—significant difference at *p* < 0.05; **—significant difference at *p* < 0.01.

**Figure 2 insects-17-00010-f002:**
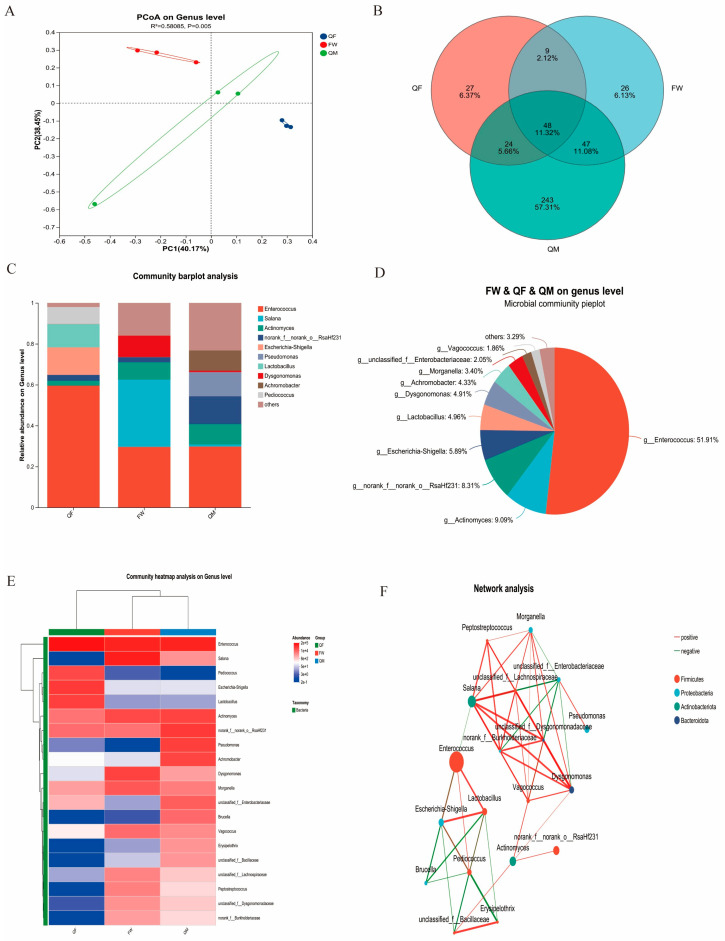
Analysis of Intestinal Microbiota in BSFL. (**A**) Principal coordinate analysis diagrams of the intestinal microbiome of BSFL cultured in different matrices based on genus level analysis. (**B**) The number of coexisting and unique species in the three groups based on the genus level analysis. (**C**) The microbial communities of the intestinal microbiota of three groups of BSFL at the genus level. (**D**) The percentage of the number of species among the total number of species. (**E**) The relative abundance of the top 20 dominant bacteria in each group. (**F**) The species correlation network diagram reflects the correlations among the top 50 species in terms of total water abundance. The size of the nodes in the figure indicates the abundance of species, and different colors represent different species. The thickness of the line indicates the magnitude of the correlation coefficient. The thicker the line, the higher the correlation between species. The more lines there are, the closer the connection between this species and other species is.

**Figure 3 insects-17-00010-f003:**
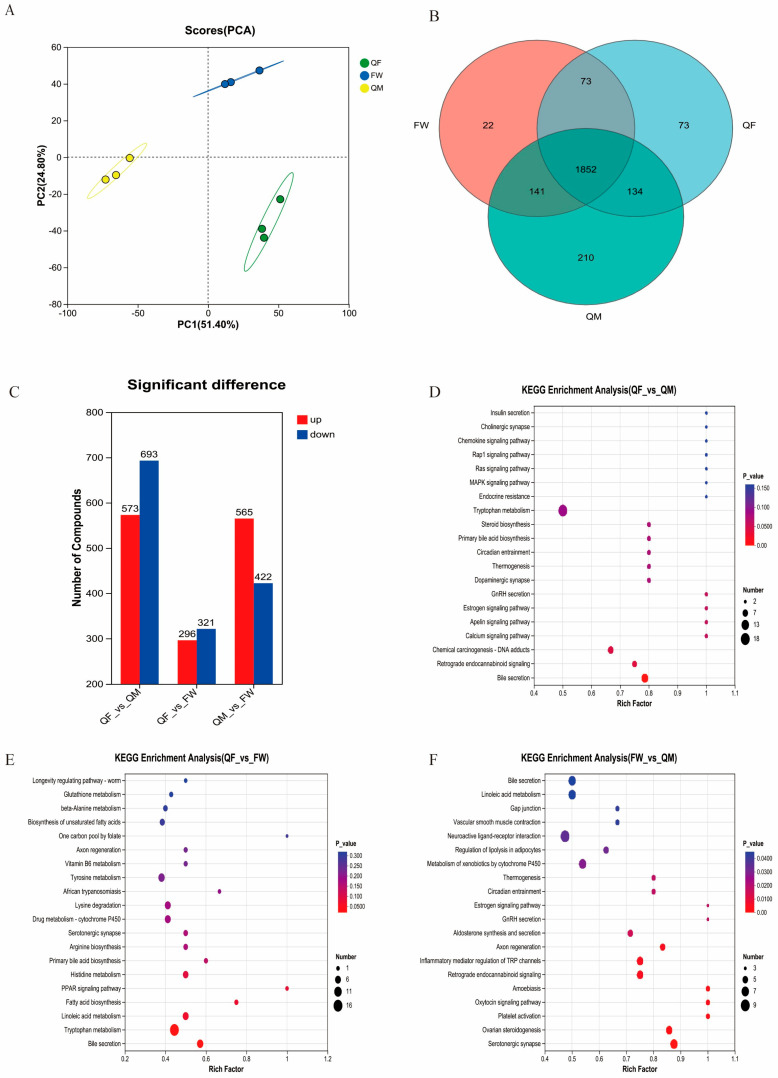
Non-targeted metabolomics analysis of LC-MS in BSFL parasites cultured in three different substrates. (**A**) The principal component analysis diagram shows the differences in the types of BSFL metabolites caused by different substrate cultures. (**B**) The venn diagram shows the common and unique types of metabolites in the three groups. (**C**) The number of metabolites with differences among the three groups. (**D**) KEGG pathways enriched with differential metabolites between the QF group and the QM group. (**E**) KEGG pathways enriched with differential metabolites between the QF group and the FW group. (**F**) KEGG pathways enriched with differential metabolites between the FW group and the QM group.

**Figure 4 insects-17-00010-f004:**
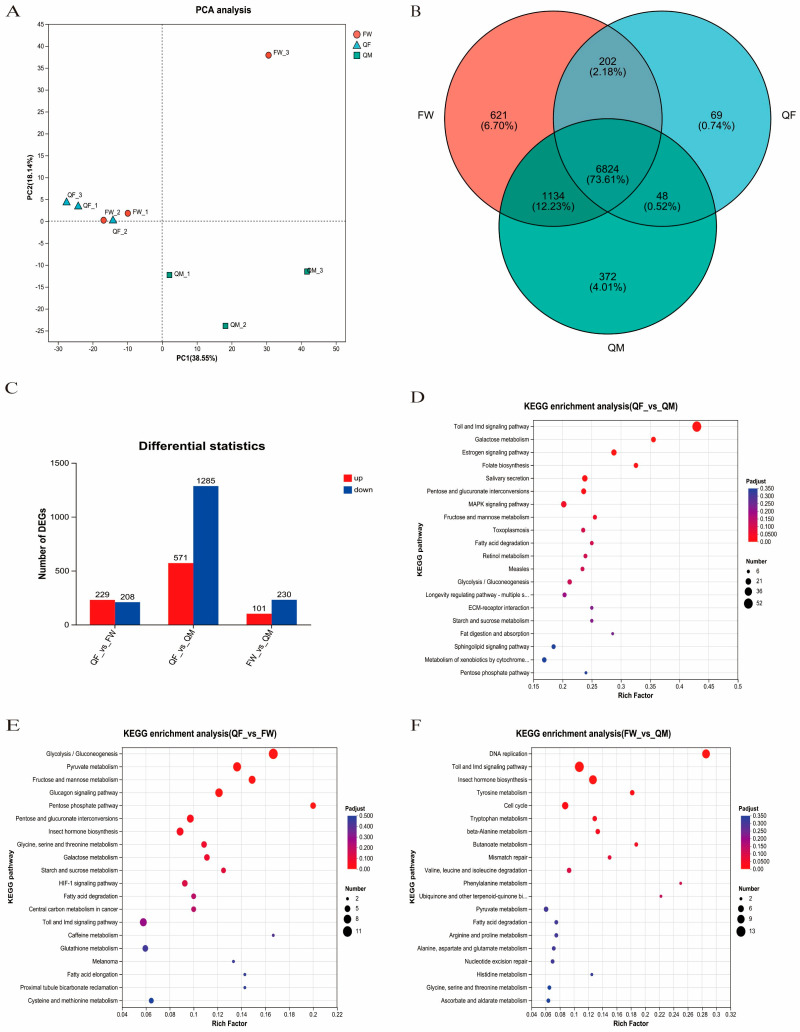
Eukaryotic parametric transcriptome analysis of BSFL parasites cultured in three different substrates. (**A**) Principal Component analysis. (**B**) Common genes and specific genes of all samples. (**C**) The number of genes with expression level differences among the three groups. (**D**) KEGG functional enrichment was performed on the genes with expression level differences between the QF group and the QM group. (**E**) KEGG functional enrichment was performed on the genes with expression level differences between the QF group and the FW group. (**F**) KEGG functional enrichment was performed on the genes with expression level differences between the FW group and the QM group.

**Table 1 insects-17-00010-t001:** Nutritional Components of Three Substrates.

Item (%DM)	QF	FW	QM	*p*-Values
Total ash	12.6 ± 0.1 ^b^	11.3 ± 0.1 ^c^	24.6 ± 0.1 ^a^	<0.001
Crude protein	21.2 ± 0.2 ^c^	35.2 ± 0.7 ^b^	41.3 ± 0.4 ^a^	<0.001
Crude fat	10.6 ± 0.3 ^a^	6.9 ± 0.1 ^a^	2.7 ± 0.1 ^b^	<0.001
Calcium	3.6 ± 0.1 ^b^	2.1 ± 0.1 ^c^	5.1 ± 0.1 ^a^	<0.001
Phosphorus	0.6 ± 0.1 ^b^	0.6 ± 0.1 ^b^	2.2 ± 0.2 ^a^	0.001

Note: QF represents quail feed, FW represents food waste, and QM represents quail manure. For peer data, different letters in the superscript labels in the same row indicate significant differences (*p* < 0.05), while the same letters indicate no significant differences (*p* > 0.05).

**Table 2 insects-17-00010-t002:** Effects of Different Substrates on the Molecular Components of BSFL.

Item (%DM)	QF	FW	QM	*p*-Values
Moisture	65.0 ± 0.6 ^c^	78.3 ± 1.3 ^a^	73.3 ± 2.0 ^b^	<0.001
Total ash	12.3 ± 0.3 ^c^	16.6 ± 0.4 ^b^	23.4 ± 0.7 ^a^	<0.001
Crude protein	37.4 ± 0.8 ^b^	47.5 ± 3.8 ^a^	48.3 ± 1.5 ^a^	<0.001
Crude fat	33.2 ± 1.3 ^a^	25.1 ± 1.1 ^b^	7.0 ± 1.1 ^c^	<0.001
Calcium	4.0 ± 0.1 ^c^	5.6 ± 0.4 ^a^	4.8 ± 0.2 ^b^	<0.001
Phosphorus	0.6± 0.1 ^a^	0.4 ± 0.1 ^b^	0.6 ± 0.1 ^a^	<0.001

Note: QF represents quail feed, FW represents food waste, and QM represents quail manure. For peer data, different letters in the superscript labels in the same row indicate significant differences (*p* < 0.05), while the same letters indicate no significant differences (*p* > 0.05).

**Table 3 insects-17-00010-t003:** Effects of Different Substrates on the Nutritional Components of BSF frass.

Item (%DM)	QF	FW	QM	*p*-Values
Moisture	45.9 ± 3.0 ^c^	85.1 ± 0.6 ^a^	66.5 ± 2.0 ^b^	<0.001
Total ash	19.8 ± 0.9 ^c^	22.2 ± 0.9 ^b^	40.6 ± 1.2 ^a^	<0.001
Crude protein	26.2 ± 0.6 ^a^	25.8 ± 2.7 ^a^	18.3 ± 0.4 ^b^	<0.001
Crude fat	2.2 ± 0.3 ^b^	3.5 ± 0.3 ^a^	2.3 ± 0.1 ^b^	0.001
Calcium	5.0 ± 0.1 ^b^	2.1 ± 0.1 ^c^	9.6 ± 0.3 ^a^	<0.001
Phosphorus	1.0 ± 0.1 ^a^	0.2 ± 0.1 ^c^	0.7 ± 0.1 ^b^	<0.001

Note: QF represents quail feed, FW represents food waste, and QM represents quail manure. For peer data, different letters in the superscript labels in the same row indicate significant differences (*p* < 0.05), while the same letters indicate no significant differences (*p* > 0.05).

**Table 4 insects-17-00010-t004:** Effects of Different Substrates on Amino Acid Content in BSFL.

Amino Acid (%DM)	QF	FW	QM	*p*-Values
Essential amino-acid	10.6 ± 0.9	11.9 ± 0.5	11.1 ± 0.5	0.809
Isoleucine	1.7 ± 0.2	1.8 ± 0.1	1.7 ± 0.2	0.413
Leucine	2.6 ± 0.3	2.8 ± 0.1	2.5 ± 0.3	0.417
Valine	2.0 ± 0.2 ^b^	2.3 ± 0.1 ^a^	2.2 ± 0.1 ^a^	0.016
Phenylalanine	1.6 ± 0.2 ^b^	2.0 ± 0.1 ^a^	1.7 ± 0.1 ^b^	0.003
Threonine	1.3 ± 0.1 ^b^	1.5 ± 0.1 ^a^	1.5 ± 0.1 ^a^	<0.001
Lysine	1.4 ± 0.2	1.6 ± 0.3	1.5 ± 0.1	0.667
Non-essential amino acid	21.4 ± 1.1 ^b^	24.0 ± 0.3 ^a^	24.4 ± 0.7 ^a^	0.002
Aspartic acid	3.7 ± 0.1 ^b^	4.4 ± 0.1 ^a^	4.3 ± 0.1 ^a^	<0.001
Serine	1.2 ± 0.1 ^b^	1.4 ± 0.1 ^a^	1.5 ± 0.1 ^a^	<0.001
Glycine	1.8 ± 0.1 ^b^	2.1 ± 0.1 ^a^	2.2 ± 0.1 ^a^	<0.001
Tyrosine	2.6 ± 0.2 ^ab^	2.8 ± 0.1 ^a^	2.2 ± 0.1 ^b^	0.009
Glutamic acid	5.1 ± 0.2 ^b^	5.4 ± 0.3 ^b^	6.2 ± 0.2 ^a^	<0.001
Histidine	1.2 ± 0.1 ^b^	1.6 ± 0.1 ^a^	1.5 ± 0.1 ^a^	0.004
Arginine	1.7 ± 0.15 ^b^	2.3 ± 0.1 ^a^	2.3 ± 0.1 ^a^	<0.001
Proline	1.7 ± 0.1	1.7 ± 0.1	1.8 ± 0.1	0.292
Alanine	2.5 ± 0.3	2.4 ± 0.1	2.5 ± 0.2	0.917

Note: QF represents quail feed, FW represents food waste, and QM represents quail manure. For peer data, different letters in the superscript labels in the same row indicate significant differences (*p* < 0.05), while the same letters indicate no significant differences (*p* > 0.05).

**Table 5 insects-17-00010-t005:** Effects of Different Substrates on Fatty Acid Content in BSFL.

Fatty Acid (% Total Identified Fatty Acids)	QF	FW	QM	*p*-Values
C10:0	1.2 ± 0.1 ^a^	1.0 ± 0.1 ^b^	0.7 ± 0.1 ^c^	0.001
C12:0	35.8 ± 1.5 ^a^	34.1 ± 2.2 ^a^	13.2 ± 0.5 ^b^	<0.001
C14:0	7.3 ± 0.7 ^a^	7.3 ± 0.7 ^a^	3.6 ±0.1 ^b^	<0.001
C16:0	15.9 ± 0.8 ^c^	20.4 ± 2.1 ^b^	29.2 ± 0.3 ^a^	<0.001
C17:0	0.1 ± 0.1 ^b^	0.2 ± 0.1 ^b^	1.2 ± 0.1 ^a^	<0.001
C20:0	16.0 ± 1.5 ^c^	22.3 ± 0.9 ^b^	31.7 ± 0.4 ^a^	<0.001
C21:0	1.0 ± 0.1 ^c^	4.3 ± 0.6 ^a^	2.9 ±0.4 ^b^	<0.001
Sum SFA	77.3 ± 0.8 ^c^	89.7 ± 0.8 ^a^	82.4 ± 1.3 ^b^	<0.001
C14:1	0.2 ± 0.1 ^b^	0.3 ± 0.1 ^b^	1.9 ± 0.3 ^a^	<0.001
C16:1n7	0.4 ± 0.1 ^c^	0.7 ± 0.1 ^b^	1.4 ± 0.1 ^a^	<0.001
C18:1n9t	3.3 ± 0.2 ^c^	4.6 ± 0.4 ^b^	6.8 ± 0.2 ^a^	<0.001
C18:2n6t	17.7 ± 0.7 ^a^	0.3 ± 0.1 ^b^	0.7 ± 0.1 ^b^	<0.001
Sum UFA	21.5 ± 0.9 ^a^	5.9 ± 0.5 ^c^	10.8 ± 0.4 ^b^	<0.001

Note: QF represents quail feed, FW represents food waste, and QM represents quail manure. For peer data, different letters in the superscript labels in the same row indicate significant differences (*p* < 0.05), while the same letters indicate no significant differences (*p* > 0.05).

**Table 6 insects-17-00010-t006:** Effects of Different Substrates on Trace Element Contents in BSFL.

Trace Element (µg/g)	QF	FW	QM	*p*-Values
Fe	353.7 ± 20.2 ^c^	588.5 ± 7.4 ^b^	707.6 ± 10.5 ^a^	<0.001
Cu	12.8 ± 1.5 ^b^	15.0 ± 1.6 ^b^	35.0 ± 0.8 ^a^	<0.001
Zn	109.1 ± 5.7 ^b^	73.8 ± 9.8 ^c^	287.6 ± 10.2 ^a^	<0.001
Mn	152.7 ± 6.4 ^b^	58.1 ± 5.1 ^c^	224.5 ± 1.4 ^a^	<0.001
K	9664.7 ± 124.2 ^b^	9653.0 ± 345.7 ^b^	21,236.5 ± 471.5 ^a^	<0.001
Na	2083.4 ± 129.5 ^c^	3752.0 ± 146.4 ^b^	4531.5 ± 91.7 ^a^	<0.001
Mg	3031.2 ± 61.0 ^b^	2292.0 ± 71.1 ^c^	4806.4 ± 113.5 ^a^	<0.001

Note: QF represents quail feed, FW represents food waste, and QM represents quail manure. For peer data, different letters in the superscript labels in the same row indicate significant differences (*p* < 0.05), while the same letters indicate no significant differences (*p* > 0.05).

## Data Availability

The data presented in this study are available on request from the corresponding author. The data are not publicly available due to privacy.

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
