# Peer review of "Effect of Rearing Substrate on Nutritional Composition, Growth Performance and Multi-Omics Characteristics of Black Soldier Fly"

_insects, 2025, doi:10.3390/insects17010010_

Round 1

Reviewer 1 Report

Comments and Suggestions for Authors

The manuscript entitled ''Effect of rearing substrate on nutritional composition, growth performance and multi-omics characteristics of black soldier fly'' is an original article that studies effects of different rearing substrates on nutritional composition, growth performance and multi-omics characteristics of black soldier fly. In general, the paper is well-written, experiments are well-conducted, and results are well-presented. Introduction should be changed a little. Specific comments are given bellow.

Simple Summary: This chapter is well-written, with only minor adjustments.

Line 115: rearing substrates

Line 17: ...of black soldier fly Hermetia illucens (Linnaeus, 1758) larvae

Line 20: The (rearing) substrates also changed...

Line 21: We conclude that food waste is...

Lines 22-23: The first part of the sentence, ’’This helps meet industry protein needs sustainably’’ does not make sense. Please, rewrite it.

Abstract: This chapter is well-written, with only minor adjustments.

Line 26: black soldier fly Hermetia illucens (Linnaeus, 1758) larvae (BSFL)

Line 29: QF-reared larvae

Line 34: Lactobacillus (in italic)

  1. Introduction: This chapter is well-written, however lacks some information. It lacks a few sentences about intestinal microbiota, what is known about BSF, or other insect-based feed organisms, and how rearing substrate changes it...

Lines 54-55: The black soldier fly (BSF),  Hermetia illucens (Linnaeus, 1758), ...

Line 65:... poultry [10,11]. – delete the full stop after [10,11]

Line 73: ...with different substrates, including poultry...

Line 74:  ...vegetables, and rotten fish,

  1. Materials and Methods: This chapter is generally well-written. However, some additional information is needed.

Line 91: …(BSFL) were evenly distributed… How evenly? How much larvae did you distribute per replication? How much of substrate did you use per replication? The number of larvae, the quantity of substrates needs to be the same in all rearing substrates, in all replications.

Lines 92-93: You presented fundamental nutritional and physical properties of the tested substrates in Table 1. How did you obtain results for these properties? If you did experimental research on this, please explain the procedure.

Line 94: As much as you can, describe food waste. Because it could be anything, based just on ’’food waste’’.

Line 96: What is a commercial feed substrate? Is it an FW? Clarify this.

Line 107: In the beginning, you must indicate that you conducted nutritional analysis of BSFL and their frass.

  1. Results: This chapter is well-written, with only minor adjustments.

Table 2: Capital P in phosphorus.

Lines 232-233, 236-237, 263-264, 267-269 and 272-273:  For peer data, different letters in the superscript labels in the same row indicate significant differences (P< 0.05), while the same letters indicated no significant differences (P> 0.05).

Line 242: significantly higher non-essential amino acid levels…

Lines 263-264: For peer data, different letters in the superscript labels in the same row indicate significant differences (P< 0.05), while the same letters or no letters indicated no significant differences (P> 0.05).

Line 296: Figure 1. Effects of three different rearing substrates…

Lines 298-299: (D) The body weight of BSFL on day 7 of collection.

Lines 299-300: * - significant difference  at P<0.05; ** - significant difference at P<0.01.

Line 323: Principal coordinates analysis...

  1. Discussion: This chapter is well-written, with only one minor adjustment.

Line 518: Enterococcus in italic

  1. Conclusion: This chapter is well-written, with only one minor adjustment.

Line 549: position and growth performance of BSF, ...

Author Response

Point-by-point Response

Dear Editor and Reviewers,

We sincerely thank you for your constructive comments and valuable suggestions on our manuscript entitled “Effect of rearing substrate on nutritional composition, growth performance and multi-omics characteristics of black soldier fly”. These comments are highly valuable and have helped us significantly improve the quality of our manuscript. We have carefully considered each point raised and have made comprehensive revisions accordingly. The point-by-point responses are detailed below.

Response to Reviewer #1

Comments and Suggestions for Authors

The manuscript entitled ''Effect of rearing substrate on nutritional composition, growth performance and multi-omics characteristics of black soldier fly'' is an original article that studies effects of different rearing substrates on nutritional composition, growth performance and multi-omics characteristics of black soldier fly. In general, the paper is well-written, experiments are well-conducted, and results are well-presented. Introduction should be changed a little. Specific comments are given bellow.

Simple Summary: This chapter is well-written, with only minor adjustments.

Comments 1: Line 15: rearing substrates

Response 1: Thank you for this constructive feedback. We have revised as suggested (line 15).

Comments 2: Line 17: ...of black soldier fly Hermetia illucens (Linnaeus, 1758) larvae

Response 2: Thank you for pointing this out. We have added the full scientific name (line 17).

Comments 3: Line 20: The (rearing) substrates also changed...

Response 3: Thank you for this constructive feedback. We have incorporated this suggestion (line 20).

Comments 4: Line 21: We conclude that food waste is...

Response 4: Thank you for this constructive feedback. The text has been modified accordingly (line 22).

Comments 5: Lines 22-23: The first part of the sentence, “This helps meet industry protein needs sustainably’’ does not make sense. Please, rewrite it.

Response 5: Thank you for highlighting this issue. We have rewritten the sentence for better clarity (line 23).

Abstract: This chapter is well-written, with only minor adjustments.

Comments 6: Line 26: black soldier fly Hermetia illucens (Linnaeus, 1758) larvae (BSFL)

Response 6: Agree. We have added the abbreviation as recommended (line 27).

Comments 7: Line 29: QF-reared larvae

Response 7: Agree. The text has been updated accordingly (line 31).

Comments 8: Line 34: Lactobacillus (in italic)

Response 8: Agree. We have corrected the formatting to italic (line 35).

Comments 9: Introduction: This chapter is well-written, however lacks some information. It lacks a few sentences about intestinal microbiota, what is known about BSF, or other insect-based feed organisms, and how rearing substrate changes it...

Response 9: We appreciate this valuable suggestion. We have added relevant information about intestinal microbiota and substrate effects on lines 75-79.

Comments 10: Lines 54-55: The black soldier fly (BSF), Hermetia illucens (Linnaeus, 1758), ...

Response 10: The formatting has been corrected as suggested (line 54-55).

Comments 11: Line 65:... poultry [10,11]. – delete the full stop after [10,11]

Response 11: Agree. We have removed the extra punctuation (line 66).

Comments 12: Line 73: ...with different substrates, including poultry...

Response 12: The text has been modified as recommended (line 79-80).

Comments 13: Line 74:  ...vegetables, and rotten fish,

Response 13: We have corrected the punctuation accordingly (line 80-81).

Materials and Methods: This chapter is generally well-written. However, some additional information is needed.

Comments 14: Line 91: …(BSFL) were evenly distributed… How evenly? How much larvae did you distribute per replication? How much of substrate did you use per replication? The number of larvae, the quantity of substrates needs to be the same in all rearing substrates, in all replications.

Response 14: Thank you for this important clarification. We have added detailed information that this experiment involved three replicates in each group, with 1,500 BSF larvae added to 1 kilogram of substrate for each replicate (lines 97-99).

Comments 15: Lines 92-93: You presented fundamental nutritional and physical properties of the tested substrates in Table 1. How did you obtain results for these properties? If you did experimental research on this, please explain the procedure.

Response 15: Thank you for pointing this out. The nutritional indicators of the substrate were also determined in accordance with the analytical method described in the materials and methods section. Samples were collected after thorough mixing of the substrate and subsequently analyzed (line 115-116).

Comments 16: Line 94: As much as you can, describe food waste. Because it could be anything, based just on “food waste’’.

Response 16: Thank you for requesting this clarification. We have added a detailed description specifying that "food waste" refers to the solid residue obtained after impurity removal and three-phase separation of kitchen waste at the treatment plant (lines 101-104).

Comments 17: Line 96: What is a commercial feed substrate? Is it an FW? Clarify this.

Response 17: Thank you for pointing this out. We have clarified that QF refers to commercially purchased quail feed (lines 100-101).

Comments 18: Line 107: In the beginning, you must indicate that you conducted nutritional analysis of BSFL and their frass.

Response 18: Thank you for pointing this out. We have added this indication at the beginning of the relevant section (lines 115-116).

Results: This chapter is well-written, with only minor adjustments.

Comments 19: Table 2: Capital P in phosphorus.

Response 19: We have corrected to "Phosphorus" as suggested (Table 2).

Comments 20: Lines 232-233, 236-237, 263-264, 267-269 and 272-273:  For peer data, different letters in the superscript labels in the same row indicate significant differences (P< 0.05), while the same letters indicated no significant differences (P> 0.05).

Response 20: We have corrected accordingly throughout the text (line 272-274, 276-278, 280-282, 317-319, 321-323, 325-327).

Comments 21: Line 242: significantly higher non-essential amino acid levels…

Response 21: Thank you for pointing this out. We have corrected this mistake (line 290-292).

Comments 22: Lines 263-264: For peer data, different letters in the superscript labels in the same row indicate significant differences (P< 0.05), while the same letters or no letters indicated no significant differences (P> 0.05).

Response 22: Thank you for pointing this out. We have revised as suggested (line 317-319).

Comments 23: Line 296: Figure 1. Effects of three different rearing substrates…

Response 23: Thank you for pointing this out. We have corrected this mistake (line 350).

Comments 24: Lines 298-299: (D) The body weight of BSFL on day 7 of collection.

Response 24: Thank you for pointing this out. We have corrected this mistake (lines 353).

Comments 25: Lines 299-300: * - significant difference at P<0.05; ** - significant difference at P<0.01.

Response 25: Agree. We have added this explanation to the figure legend (lines 353-354).

Comments 26: Line 323: Principal coordinates analysis...

Response 26: The terminology has been corrected (lines 377).

Discussion: This chapter is well-written, with only one minor adjustment.

Comments 27: Line 518: Enterococcus in italic

Response 27: We have formatted Enterococcus in italic as recommended (lines 571).

Conclusion: This chapter is well-written, with only one minor adjustment.

Comments 28: Line 549: position and growth performance of BSF, ...

Response 28: The wording has been corrected accordingly (lines 598).

Reviewer 2 Report

Comments and Suggestions for Authors

The methods applied are not scientifically sound or lack so much information that this study cannot be replicated by any means. Information that is, for example lacking, is the number of larvae per replicate container, number of replicate containers per treatment group, methods used for GC-MS analysis, how samples were treated before nutritional analysis, substrate information including dry matter content, amino acid profile, fatty acid profile, and minerals. 

Author Response

Point-by-point Response

Dear Editor and Reviewers,

We sincerely thank you for your constructive comments and valuable suggestions on our manuscript entitled “Effect of rearing substrate on nutritional composition, growth performance and multi-omics characteristics of black soldier fly”. These comments are highly valuable and have helped us significantly improve the quality of our manuscript. We have carefully considered each point raised and have made comprehensive revisions accordingly. The point-by-point responses are detailed below.

Reviewer 2

Comments: The methods applied are not scientifically sound or lack so much information that this study cannot be replicated by any means. Information that is, for example lacking, is the number of larvae per replicate container, number of replicate containers per treatment group, methods used for GC-MS analysis, how samples were treated before nutritional analysis, substrate information including dry matter content, amino acid profile, fatty acid profile, and minerals. 

Response: We sincerely thank the reviewer for these important comments and valuable suggestions regarding methodological details. We appreciate the opportunity to improve our manuscript and have thoroughly revised it to address these concerns, ensuring the reproducibility of our study. The specific revisions are as follows:

  1. We have improved the Materials and Methods section and specified the experimental design details, stating that there were three replicates per treatment group, with each replicate containing 1,500 BSF larvae reared on 1 kg of substrate (Lines 98-99).
  2. We have supplemented the detailed procedures for amino acids (HPLC, line 128-133), fatty acids (GC, line 145-155) and metabolome (MS, line 209-215) analysis, including sample preparation, instrument parameters, and data processing methods.
  3. We have added clear descriptions of the sample handling procedures prior to nutritional analysis, specifically that larvae were immediately frozen in liquid nitrogen after collection and stored at -20°C until analysis (Lines 113-116).
  4. We have provided a detailed explanation of the source and definition of the substrate (line 100-104).

Reviewer 3 Report

Comments and Suggestions for Authors

This paper describes a wide selection of experimental results from BSF larvae reared on 3 different feed substrates. It is a very detailed study on many aspects of BFS larvae reared on the different substrates. It is, however, highly descriptive. Results are to some extent compared to a not complete selection of results from the literature, but the results are hardly discussed. Most of what is presented has been investigated before. It would have been good if the authors had introduced their results into relevant context of current knowledge, in order to pinpoint potential novel findings. I did not really figure out if this paper presents any new knowledge or insights. Alternatively, which may also be fine, the authors could make it clearer that the value of this study is the repetition of former studies (maybe on novel feed substrates), but here in more details than normally seen. This could justify a descriptive study, I think, because the data seems solid. For these reasons, I recommend that the authors revised the manuscript. Below, I added specific comments and questions.

Introduction

In general, I think the authors use too much space on subjects only marginally relevant to the this study, and at the same time mis not to introduce current knowledge from the many prior studies that are similar to this one.

Line 41-52. It is pretty hard to see these general considerations as really relevant for the present study?

Line 48. As far as I am informed, the human population no longer increases exponentially?

Line 60. Why ‘high-quality proteins’ and not just ‘proteins’?

Line 70-72. Are you sure that it is not mostly the fat content that is variable and affected by the feed substrate? If fat content is high, other components will be comparatively low.

Line 80-84. Many studies have evaluated growth, molecular composition, and to some extent also microbiome of BSF larvae on a wide selection of different feed substrates. I suggest you specify what may be new in this study.

Methods and materials

Table 1. Can be improved

-Legend. Unclear what you mean by indicators?

-Why not just describe the content as e.g. substrate components?

-What is the unit - % of what (e.g. wet or dry weight)?.

-Please reconsider your use of significant digits (throughout the manuscript). If accuracy is 0.1%, there is no point in reporting data  with 0.01% accuracy.

-In the note, what do you mean be 'group'? And the differences mentioned, these are differences between what?

-And lastly, if these data are the results of analyses, I think Table 1 belongs in the Results section.

Line 150-157. It is, of course, a matter of preference, but it would be more elegant to use symbols instead of text in equations.

Results

Table 2. Similar comments as for Table 1 (and for the following tables as well).

-It is strange to name the components that build up the larvae, nutritional indicators. Why not just e.g. molecular components?

-Unit is % of what?

-Please reconsider the use of significant digits

-Can you explain the high ash content of QM-larvae, when the ash apparently do not represent higher contents of calcium? Can these results be trusted?

-As seen before, high lipid is associated to low moisture content in the BSF larvae (see Eriksen (2022) PLoS ONE 17: e0276605, https://doi.org/10.1371/journal.pone.0276605). High fat must also be contributing to the low protein content. I suggest this is taken into consideration in your discussion of your results.

Figure 1. Strange to name length and weight 'growth indicators'. Why don't you just use 'length and weight'?

Discussion

The Discussion section is quite low on discussion. It is mainly introductory information that would fit better into the Introduction section, repetition of the results, and some comparisons but without really discussing what it all means. I think it weakens the paper, that you don't introduce former knowledge on how feed composition affects BSF larvae, and use this information to pinpoint if you found out something new. I may be wrong, but it seems to me that you have done a detailed study, but maybe not a very original one, and therefore you also did not really find out anything new. But, you have made a detailed experimental characterization of BSF larvae on 3 substrates, maybe this is the real achievement? But I think you should tell this more clearly.

Line 382-398. It should not be needed to start the Discussion by a second introduction. It is better to keep all introduction in the Introduction section. Furthermore, it is also a bit strange here to focus on aquaculture when aquaculture was not mentioned in the Introduction section.

Line 398-405. Please note that you are merely repeating your results but not really discussing them. The same goes for the sections below.

Line 406-432. The amino acid composition of BSF larvae has also been described many times before. I think it is a mistake not to use these former results to put your results into a relevant perspective and pinpoint if you found out anything new. For now, it is not clear to me what you tried to achieve doing the amino acid analysis, which has not been done before?

Line 433-442. Also numerous analyses of fatty acid composition in BSF larvae can be found in the literature.

Line 512-513. What do you mean by ‘quail manure exerted the strongest influence on BSFL metabolism’, and what is the evidence for this?

Line 532. What do you mean by 'systematically'?

Author Response

Point-by-point Response

Dear Editor and Reviewers,

We sincerely thank you for your constructive comments and valuable suggestions on our manuscript entitled “Effect of rearing substrate on nutritional composition, growth performance and multi-omics characteristics of black soldier fly”. These comments are highly valuable and have helped us significantly improve the quality of our manuscript. We have carefully considered each point raised and have made comprehensive revisions accordingly. The point-by-point responses are detailed below.

Reviewer 3

This paper describes a wide selection of experimental results from BSF larvae reared on 3 different feed substrates. It is a very detailed study on many aspects of BFS larvae reared on the different substrates. It is, however, highly descriptive. Results are to some extent compared to a not complete selection of results from the literature, but the results are hardly discussed. Most of what is presented has been investigated before. It would have been good if the authors had introduced their results into relevant context of current knowledge, in order to pinpoint potential novel findings. I did not really figure out if this paper presents any new knowledge or insights. Alternatively, which may also be fine, the authors could make it clearer that the value of this study is the repetition of former studies (maybe on novel feed substrates), but here in more details than normally seen. This could justify a descriptive study, I think, because the data seems solid. For these reasons, I recommend that the authors revised the manuscript. Below, I added specific comments and questions.

Introduction

In general, I think the authors use too much space on subjects only marginally relevant to this study, and at the same time mis not to introduce current knowledge from the many prior studies that are similar to this one.

Comments 1: Line 41-52. It is pretty hard to see these general considerations as really relevant for the present study?

Response 1: Thank you for pointing this out. These contents form the broad context of this research, and we believe it is of great significance for readers to understand the meaning and background of this research. It is precisely because of the backdrop of food waste and feed shortage that BSF, with its strong ability to convert and utilize waste, has become a research focus. Meanwhile, we have streamlined the relevant content as you suggested (line 48-52).

Comments 2: Line 48. As far as I am informed, the human population no longer increases exponentially?

Response 2: Thank you for this correction. We have revised the relevant content to be more accurate (Lines 48-52).

Comments 3: Line 60. Why ‘high-quality proteins’ and not just ‘proteins’?

Response 3: Thank you for pointing this out. We want to emphasize the high quality of BSF protein (such as balanced amino acids), but for better understanding, we have modified the term to 'proteins' as suggested for better precision (Line 61).

Comments 4: Line 70-72. Are you sure that it is not mostly the fat content that is variable and affected by the feed substrate? If fat content is high, other components will be comparatively low.

Response 4: Thank you for this important insight. We have revised our description to acknowledge that fat content variability significantly affects other components and have incorporated this perspective throughout our results interpretation (Lines 71-75).

Comments 5: Line 80-84. Many studies have evaluated growth, molecular composition, and to some extent also microbiome of BSF larvae on a wide selection of different feed substrates. I suggest you specify what may be new in this study.

Response 5: Thank you for this valuable suggestion. We have clarified the novel aspects of our study, emphasizing the systematic comparison of common substrates using integrated nutritional, transcriptomic, and metabolomic approaches, which provides unprecedented insights into the molecular mechanisms underlying substrate-dependent variations (Lines 86-89).

Methods and materials

Table 1. Can be improved

Comments 6: Legend. Unclear what you mean by indicators?

Response 6: We have changed the title to “Nutritional Components of Three Substrates” (Lines 271).

Comments 7: Why not just describe the content as e.g. substrate components?

Response 7: We have changed the title to “Nutritional Components of Three Substrates” (Lines 271).

Comments 8: What is the unit - % of what (e.g. wet or dry weight)?

Response 8: Thank you for pointing this out. We have specified that values are expressed as % dry matter (DM).

Comments 9: Please reconsider your use of significant digits (throughout the manuscript). If accuracy is 0.1%, there is no point in reporting data with 0.01% accuracy.

Response 9: We have adjusted significant digits to 0.1% throughout the manuscript as you suggested.

Comments 10: In the note, what do you mean be 'group'? And the differences mentioned, these are differences between what?

Response 10: Thank you for raising this point. We have clarified in the table notes that the statistical comparison is based on the differences among various rearing substrates (line 272).

Comments 11: And lastly, if these data are the results of analyses, I think Table 1 belongs in the Results section.

Response 11: Thanks for your suggestion. We have moved Table 1 to the Results section (line 244-253; 271-274).

Comments 12: Line 150-157. It is, of course, a matter of preference, but it would be more elegant to use symbols instead of text in equations.

Response 12: We appreciate this suggestion and we have revised accordingly in the manuscript (line 174-183).

Results

Table 2. Similar comments as for Table 1 (and for the following tables as well).

Comments 13: It is strange to name the components that build up the larvae, nutritional indicators. Why not just e.g. molecular components?

Response 13: Agree. We have revised accordingly in the manuscript (line 275).

Comments 14: Unit is % of what?

Response 14: Specified that values are expressed as % dry matter (DM).

Comments 15: Please reconsider the use of significant digits

Response 15: Adjusted significant digits to 0.1% throughout the manuscript.

Comments 16: Can you explain the high ash content of QM-larvae, when the ash apparently do not represent higher contents of calcium? Can these results be trusted?

Response 16: As suggested, the high ash content in QM-larvae without corresponding high calcium levels can be explained by: (1) the complex composition of quail manure potentially inhibiting calcium absorption, as evidenced by high calcium content in the substrate and frass but not the insect; (2) the high ash content of QM-larvae is not due to high calcium content, but rather because insects accumulate more ash in their bodies when using substrates with inherently low carbohydrate and high mineral content such as poultry manure; and (3) the significant increase in other mineral elements in QM-larvae (Table 6) contributing to ash content while diluting the relative calcium percentage.

Comments 17: As seen before, high lipid is associated to low moisture content in the BSF larvae (see Eriksen (2022) PLoS ONE 17: e0276605, https://doi.org/10.1371/journal.pone.0276605). High fat must also be contributing to the low protein content. I suggest this is taken into consideration in your discussion of your results.

Response 17: We appreciate this insightful suggestion. We have expanded our explanation in the discussion section to address this apparent discrepancy (lines 452-458).

Comments 18: Figure 1. Strange to name length and weight 'growth indicators'. Why don't you just use 'length and weight'?

Response 18: We appreciate this suggestion and we have revised accordingly in the manuscript (line 350).

Discussion

The Discussion section is quite low on discussion. It is mainly introductory information that would fit better into the Introduction section, repetition of the results, and some comparisons but without really discussing what it all means. I think it weakens the paper, that you don't introduce former knowledge on how feed composition affects BSF larvae, and use this information to pinpoint if you found out something new. I may be wrong, but it seems to me that you have done a detailed study, but maybe not a very original one, and therefore you also did not really find out anything new. But, you have made a detailed experimental characterization of BSF larvae on 3 substrates, maybe this is the real achievement? But I think you should tell this more clearly.

Response: Thank you so much for acknowledging the detailed nature of our experimental characterization. We have now significantly strengthened the discussion section by:

  1. Integrating our findings more comprehensively with current literature.
  2. Restructuring the discussion to focus on interpretation rather than just repetition of results.
  3. Explicitly stating the novel aspects of our comprehensive multi-omics approach.

Comments 19: Line 382-398. It should not be needed to start the Discussion by a second introduction. It is better to keep all introduction in the Introduction section. Furthermore, it is also a bit strange here to focus on aquaculture when aquaculture was not mentioned in the Introduction section.

Response 19: Thank you for pointing this out. We have removed the redundant introductory material and integrated relevant contextual information into the Introduction section (line 436-442).

Comments 20: Line 398-405. Please note that you are merely repeating your results but not really discussing them. The same goes for the sections below.

Response 20: Thank you for pointing this out. We have substantially revised these sections to provide proper interpretation and discussion of results in the context of existing literature (line 452-458).

Comments 21: Line 406-432. The amino acid composition of BSF larvae has also been described many times before. I think it is a mistake not to use these former results to put your results into a relevant perspective and pinpoint if you found out anything new. For now, it is not clear to me what you tried to achieve doing the amino acid analysis, which has not been done before?

Response 21: We sincerely thank the reviewer for this important suggestion. Specifically, our amino acid profiling served as one key piece of evidence within our integrated multi-omics approach, demonstrating that the food waste (FW) substrate produced larvae with a more balanced and favorable amino acid profile compared to the other substrates tested. We have revised the relevant section to better contextualize our amino acid results within existing literature and to more clearly state their role in supporting our overall conclusions about substrate selection (Lines 459-474).

Comments 22: Line 433-442. Also numerous analyses of fatty acid composition in BSF larvae can be found in the literature.

Response 22: We thank the reviewer for this pertinent observation. Rather, the fatty acid analysis served as a crucial component of the comprehensive evaluation. Specifically, it provided key evidence that the food waste (FW) substrate supported the accumulation of a more desirable fatty acid profile in the larvae, we have revised the corresponding section to better frame our fatty acid results within the context of prior knowledge and to more clearly articulate their role in the integrated assessment of substrate quality, aligning with the overall narrative of the manuscript (Lines 477-489).

Comments 23: Line 512-513. What do you mean by ‘quail manure exerted the strongest influence on BSFL metabolism’, and what is the evidence for this?

Response 23: Thank you for pointing this out. We have clarified that this conclusion is based on the QM-larvae showing the highest number of significantly altered metabolites in our metabolomic analysis, indicating the most substantial metabolic reprogramming among the three substrates (line 558-564).

Comments 24: Line 532. What do you mean by 'systematically'?

Response 24: Thank you for pointing this out. This is a word usage error and we have removed it.

Round 2

Reviewer 2 Report

Comments and Suggestions for Authors

This has been the best revised version of a manuscript that I have seen, the authors have improved the manuscript a lot. There are some minor comments that need to be addressed, that I have commented again in the manuscript. 

Author Response

Response to Reviewer 2

We sincerely thank you for the thorough review and the valuable comments and suggestions provided. We have carefully considered each point and revised the manuscript accordingly. Please find our point-by-point responses below.

Removals

We have removed the phrases and sentences as suggested:

  • Line 17: "which are a sustainable protein source"
  • Line 21: "and how their bodies process nutrients."
  • Line 23: "supports cheaper, greener feed for livestock,"
  • Line 37: "and cost-effective production"
  • Line 452: "Larvae from FW presented a 'high-protein, high-fat' profile, a hallmark of nutrient-dense organic waste."
  • Line 459-460: "Amino acid analysis revealed that glutamic acid was the most abundant amino acid across all groups, underscoring its critical role in energy metabolism and neural function."
  • Line 483-484: "The abundance of lauric acid, known for its antioxidant and hepatoprotective properties, further enhances the health-promoting value of BSFL as a feed ingredient [42]."
  • Line 487: "like lauric acid"
  • Line 589: "cost-effectiveness."

Modifications

We have modified the specified lines as recommended (marked with red font):

  • Line 18: Changed to "higher protein content, faster growth,"
  • Line 19: Changed to "higher in fat; while larvae reared on quail manure had the highest total mineral content......"
  • Line 21: Changed to "most suitable tested"
  • Line 23: Changed to "raw materials high in protein"
  • Line 38: Changed to "high larval protein contents"
  • Line 71: Changed to "An optimal C/N ratio in the substrate"
  • Line 126: Changed to "Proximate analyses were"
  • Line 270-271: Changed to "Crude fat content was highest in the QF group (33.2 ± 1.3%), followed by the FW group (25.1 ± 1.1%), and lowest in the QM group (7.0 ± 1.1%),"
  • Line 277: Changed to "highest"
  • Line 330: Added "% total identified fatty acids" to the table note (Table 5)
  • Line 455-457: Modified to "On the other hand, larvae reared on QM had high contents of protein and ash, with low lipids, characteristic of larvae reared on manure-based substrates, i.e. low carbohydrate and high mineral content [15]."
  • Line 460-461: Modified to "Larvae reared on FW and QM had higher contents of several essential amino acids,"
  • Line 554: Changed to "an essential amino acid,"
  • Line 580: Changed to "larval lipids"
  • Line 588: Changed to "among the tested substrates,"
  • Line 591: Changed to "in the larvae"

Line 31 Are you talking here about the substrate or the larvae reared on these substrates?

Response: Agreed. We have clarified the comparison objects in the abstract. The text now reads: "FW and QM groups achieved higher crude protein than the QF group" and "QF-reared larvae contained more fat than the other groups" (Line 30-31).

Line 107 “In contrast, no additional water was added to the FW and QM substrates, as their natural moisture content was considered adequate to support larval growth.” Add reference

Response: A relevant citation has been added to support this statement (reference 24) (line 117).

Line 110 “monitored daily” How?

Response: We have clarified the method. The sentence has been revised to indicate that monitoring was based on the "growth index (larval weight and length)" (line 120).

Line 167 Do you think 20/1500 is representative? Did you base this number on calculations or a previous publication? Same goes for the length determination

Response: The sample size for larval measurements (20 out of 1500) was determined based on established methods described in previous literature. A citation has been added to justify this sampling approach (reference 24). Meanwhile, we will also consider using larger sample sizes in future experiments.

Line 179 “SWw: wet weight of substrate.” Including the larvae?

Response: This refers specifically to the initial weight of the substrate added (1 kg), excluding the larvae. The text has been checked for clarity (line 108).

Line 466 Glutamic acid is not considered limiting

Response: Agreed. Glutamic acid has been removed from the discussion in this paragraph.

Line 467-469 you can’t conclude that only based on the leucine/isoleucine ratio

Response: We agree with this assessment. The corresponding sentence has been removed from the manuscript.

Line 479-480 This reference does not show antimicrobial or antiviral activities, nor that it can be converted into monolaurin in vivo nor that it can act as a natural alternative to antibiotics in feed. Generally, the consensus is that the high content of saturated fatty acids in BSF is limiting their applications in feed (St-Hilaire et al., 2007; Surendra et al., 2016; Ewald et al., 2020)

Response: Thank you for this important correction. The statement has been completely rewritten to more accurately reflect the established role of lauric acid in BSFL and its potential implications, citing more appropriate literature. The claim about it being a natural alternative to antibiotics has been removed (line 475-478).

Line 509-516 You cannot really conclude this based on the current study, as you did not run a trial with the same diet and different lipid contents, many other factors can impact growth such as the substrate waterholding capacity, pH, temperature, etc.

Response: We agree that the stated conclusion was too strong based on our experimental design. This entire paragraph has been removed from the Discussion section.

Reviewer 3 Report

Comments and Suggestions for Authors

The paper has been improved. The authors now put more emphasis on discussing. However, I still find it difficult to extract if they made new discoveries beside performing a very detailed characterization of many aspects of BFS larvae reared on 3 selected substrates. More knowledge exists on how feed substrates affect BSF larvae than account for in the paper. Below, I added specific comments and questions.

I am also not sure if the authors use the literature properly. I looked up 3 references in the manuscript, of which 2 seem to be off topic. I suggest the authors check their list of references.

Abstract

Line 30 and 31 and elsewhere. When you write ‘higher’ and ‘more’, I think you have to describe higher than what, or more than what?

Line 33. A yield will normally describe weigh  of larvae obtained per weigh of substrate used. You only describe larval weight (and since you don’t say how much substrate was used, the information is not very informative. I also think you should reevaluate your use of significant digits.

Line 36. How do you know FW is optimal – meaning no other feed substrates will be better?

Line 36 (and Line 589). I cannot see you analyze the economy in the paper, so maybe conclusions on cost-efficiency are premature?

Line 37. Unclear what you mean by ‘tailored’? You tested 3 feed substrates, but did not try to tailor anything.

Introduction

Line 79. Paper 17 does not investigate cellulose or lignin degradation in BSF cultures but effects of particle size?

Line 79. ‘Similar’ to what?

Line 84-85. Please justify your choice of quail feed.

Line 86-90. Please explain you choices of feed substrates, and explain why you expect to learn something new, which has not already been learned from using the substrates mentioned in Line 79-82. I suggest you reconsider if you make the best use of the Introduction section. In Line 42-68, you introduce some general aspects only marginally related to your study, while you don't introduce current knowledge and only a few of the relevant studies on how substrates affect growth, composition and microbiome of BSF larvae. Without such introduction, it is difficult for readers (at least for me) to find out if your results provide new discoveries. With respect to your aim, I think your paper may provide a foundation for future practical applications, but I don't think you provide a scientific basis for future research. For this, you must give the interpretation of your results more attention in the context of current knowledge.

Results

Text on figures is small and sometimes difficult to read.

Discussion

Line 436-447. There is no discussion in this text. It is Introduction. Maybe it is misplaced?

Line 438. Ref 14 is about the use of house fly maggots in aquaculture. Is this the right reference?

Line 438-439. Many studies have investigated various aspects of how substrates affect growth, composition, and microbiome of BSF larvae. Since you have not introduced these studies, it is difficult to figure out if you found out something new.

Conclusions

Line 608-609. For now, I don't agree to the word 'scientific', because you don't offer mechanistic or theoretical explanations for your results. Instead, I find your study may offer an empirical basis for substrate selection.

Author Response

Response to Reviewer 3

Comments and Suggestions for Authors

The paper has been improved. The authors now put more emphasis on discussing. However, I still find it difficult to extract if they made new discoveries beside performing a very detailed characterization of many aspects of BFS larvae reared on 3 selected substrates. More knowledge exists on how feed substrates affect BSF larvae than account for in the paper. Below, I added specific comments and questions.

I am also not sure if the authors use the literature properly. I looked up 3 references in the manuscript, of which 2 seem to be off topic. I suggest the authors check their list of references.

Response: Thank you for this valuable suggestion. We have carefully re-checked all references in the manuscript to ensure their relevance and accuracy. Inappropriate citations have been replaced or removed. 

Abstract

Line 30 and 31 and elsewhere. When you write ‘higher’ and ‘more’, I think you have to describe higher than what, or more than what?

Response: Agreed. We have clarified the comparison objects in the abstract. The text now reads: "FW and QM groups achieved higher crude protein than the QF group" and "QF-reared larvae contained more fat than the other groups" (Line 30-31).

Line 33. A yield will normally describe weigh of larvae obtained per weigh of substrate used. You only describe larval weight (and since you don’t say how much substrate was used, the information is not very informative. I also think you should reevaluate your use of significant digits.

Response: We have modified the unit to g/kg (Line 33) to better represent the yield. And it is stated in the material and method that each group consists of 3 replicates, with 1,500 BSF larvae added to 1 kilogram of substrate for each replicate (line 107-109). We have also re-evaluated the use of significant digits throughout the manuscript.

Line 36. How do you know FW is optimal – meaning no other feed substrates will be better?

Response: We agree that "optimal" might be too strong. We have modified this to state that FW was the "most suitable among the three substrates tested" based on a comprehensive evaluation of larval nutrition and growth performance indicators (Line 37 and 588).

Line 36 (and Line 589). I cannot see you analyze the economy in the paper, so maybe conclusions on cost-efficiency are premature?

Response: We agree. The term "cost-effective" has been removed from the abstract and conclusion sections.

Line 37. Unclear what you mean by ‘tailored’? You tested 3 feed substrates, but did not try to tailor anything.

Response: We have rephrased this sentence for clarity. The intended meaning was that FW is the most suitable substrate for BSFL production among those tested. The text has been revised accordingly (Line 36-38).

Introduction

Line 79. Paper 17 does not investigate cellulose or lignin degradation in BSF cultures but effects of particle size?

Response: Thank you for pointing this out. We have revised this paragraph. The main point was to illustrate how substrates influence the BSFL gut microbiome and its functions. While Reference 17 primarily investigated particle size, it also discussed how substrates affect the BSFL gut microbiome and the role of microbial communities (Line 74-77).

Line 79. ‘Similar’ to what?

Response: We have replaced "Similar" with "Studies have shown" for better clarity (Line 77).

Line 84-85. Please justify your choice of quail feed.

Response: We have expanded the justification for our substrate choices in the Introduction (line 80-86). Our previous research involved the application of BSFL in quail diets. To comprehensively assess the value of BSFL in quail production systems, we investigated the utilization of both quail feed (QF) and quail manure (QM) by BSFL. While studies exist on BSFL utilization of chicken manure, research on quail manure is scarce. Quail manure, derived from high-protein diets, potentially offers higher nutritional value. QF was chosen as a nutritionally consistent reference substrate (compared to the variability of food waste and manure), and it represents a potential waste stream (e.g., spilled feed) in quail farms.

Line 86-90. Please explain you choices of feed substrates, and explain why you expect to learn something new, which has not already been learned from using the substrates mentioned in Line 79-82. I suggest you reconsider if you make the best use of the Introduction section. In Line 42-68, you introduce some general aspects only marginally related to your study, while you don't introduce current knowledge and only a few of the relevant studies on how substrates affect growth, composition and microbiome of BSF larvae. Without such introduction, it is difficult for readers (at least for me) to find out if your results provide new discoveries. With respect to your aim, I think your paper may provide a foundation for future practical applications, but I don't think you provide a scientific basis for future research. For this, you must give the interpretation of your results more attention in the context of current knowledge.

Response: We studied the application of BSF live insects in quails (Liu, K.; Zhang, G.; Li, Y.; Jiao, M.; Guo, J.; Shi, H.; Ji, X.; Zhang, W.; Quan, K.; Xia, W. (2025). Effects of feeding unprocessed whole black soldier fly (Hermetia illucens) larvae on performance, biochemical profile, health status, egg quality, microbiome and metabolome patterns of quails. Poultry Science, 104(9), 105374. https://doi.org/10.1016/j.psj.2025.105374). Then, in order to have a more comprehensive understanding of the comprehensive application value of BSF in quail production, we conducted research on the utilization of BSF in quail manure. Currently, there are some studies on the utilization of BSF in the manure of broilers and laying hens, research on quail manure is scarce. Since the protein level of quail feed is higher than that of broilers and laying hens, the nutritional content in quail manure is also potentially higher. On one hand, quail feed has a relatively clear and consistent nutritional value. On the other hand, in quail farms, sometimes there are some wasted feeds (such as spilled feed, etc.), which can also be utilized by insects. Therefore, we chose quail feed and quail manure for comparison, and also selected kitchen waste, which is currently the most commonly used substrate for raising BSF, for comparison. Furthermore, a key novelty of our study lies in the application of multi-omics (transcriptomics and metabolomics alongside microbiome analysis) to elucidate the molecular mechanisms underlying BSFL utilization of these different substrates, which is currently underexplored, especially for quail feed and quail manure (line 80-95).

Results

Text on figures is small and sometimes difficult to read.

Response: We apologize for this issue. We have increased the font size in all figures, especially in Figure 1F (production performance indicators) and the omics figures, to ensure they are clear and legible.

Discussion

Line 436-447. There is no discussion in this text. It is Introduction. Maybe it is misplaced?

Response: We agree. This paragraph, which was more introductory in nature, has been removed from the Discussion section.

Line 438. Ref 14 is about the use of house fly maggots in aquaculture. Is this the right reference?

Response: Thank you for pointing this out. This reference has been replaced.

Line 438-439. Many studies have investigated various aspects of how substrates affect growth, composition, and microbiome of BSF larvae. Since you have not introduced these studies, it is difficult to figure out if you found out something new.

Response: Thank you for this important feedback. We acknowledge this point. As mentioned in the response regarding the Introduction, we have now incorporated more relevant literature into the Introduction and Discussion sections, particularly focusing on studies related to substrate effects and the current state of omics research in BSFL. We emphasize that while microbial community studies are more prevalent, concurrent transcriptomic and metabolomic analyses comparing different common substrates (like FW and quail manure) are lacking. Our multi-omics approach aims to provide a deeper understanding of the molecular mechanisms involved in substrate utilization by BSFL (line88-95, 574-576).

Conclusions

Line 608-609. For now, I don't agree to the word 'scientific', because you don't offer mechanistic or theoretical explanations for your results. Instead, I find your study may offer an empirical basis for substrate selection.

Response: Agreed. We have modified the conclusion accordingly, removing the word 'scientific' and rephrasing to state that our study provides an "empirical basis" for substrate selection (Line 99, 598-599).

Round 3

Reviewer 3 Report

Comments and Suggestions for Authors

The paper is being improved stepwise, and the authors have addressed the major points of criticism that I have previously raised. I still find the paper has some weaknesses, especially with respect to pinpointing what novel discoveries that may have been made in this study. The study still lacks a good theoretical foundation and a reasonable account for the existing knowledge. Still, it is a highly detailed characterization of many aspects of BSF production on 3 selected feed substrates. One suggestion could be to make it clearer that this paper represents a broad empirical dataset on BSF larvae reared on 3 different feeds.

Line 85 is one example of missing theoretical background. It is now described ‘While studies exist on BSFL utilization of chicken manure, research on QM is scarce’, but there is still no explanation on why and how chicken and quail manure differs? Former studies in BSF performances on different feeds, besides some very general comments on composition and size, are also still not included. Therefore, it will be difficult for readers to put the variables calculated from the equations in Line 186-190 into a relevant context.

Line 76-79 is an example of, in my mind, insufficient introduction of current knowledge. It has now been described ‘It is worth noting that substrates can influence the intestinal microbiota of BSF, these microbiotas can regulate the nutritional conversion efficiency of BSF by participating in the decomposition of complex organic matter and the synthesis of bioactive substances’, but nothing about how the microbiota is influenced, which substrates, which microbes, or what it means to the BSF larvae.

Still, Line 42-69 is only marginally important to this study. Thus, this space could be used better.

As mentioned above, this is a highly detailed characterization of many aspects of BSF production on 3 selected feed substrates, and it may provide useful empirical information. Of course, it is up to the authors to decide how to frame their paper. I just hope my comments have been somewhat useful, despite being a bit critical. I cannot help recommending the authors take the above comments into consideration, thus I must recommend minor revisions once more.  

Author Response

Response to Reviewer 3

We sincerely thank you for your continued engagement and for providing further constructive feedback, which has been invaluable in refining our manuscript. We have carefully considered the remaining comments and have made further revisions to strengthen the theoretical foundation, contextualize our findings within existing knowledge, and clarify the nature and contribution of our work. Our point-by-point responses and the corresponding revisions are detailed below.

General Comment:

The paper is being improved stepwise, and the authors have addressed the major points of criticism that I have previously raised. I still find the paper has some weaknesses, especially with respect to pinpointing what novel discoveries that may have been made in this study. The study still lacks a good theoretical foundation and a reasonable account for the existing knowledge. Still, it is a highly detailed characterization of many aspects of BSF production on 3 selected feed substrates. One suggestion could be to make it clearer that this paper represents a broad empirical dataset on BSF larvae reared on 3 different feeds.

Response: Thank you for this thoughtful summary and constructive suggestion. We agree that framing our study as a comprehensive empirical characterization effectively communicates its scope and utility. We have revised the paper especially Introduction section to more clearly position the manuscript as providing a detailed, multi-faceted empirical dataset on BSFL reared on three distinct substrates – food waste (FW), quail feed (QF), and quail manure (QM). We have also worked to strengthen the theoretical background and better integrate existing knowledge, as detailed in the specific responses below (marked with blue shadow).

Line 85 is one example of missing theoretical background. It is now described 'While studies exist on BSFL utilization of chicken manure, research on QM is scarce', but there is still no explanation on why and how chicken and quail manure differs? Former studies in BSF performances on different feeds, besides some very general comments on composition and size, are also still not included. Therefore, it will be difficult for readers to put the variables calculated from the equations in Line 186-190 into a relevant context.

Response: We appreciate your valuable suggestion. We have significantly expanded the justification in the Introduction (line 87-94). The revision now explicitly discusses the key compositional differences between quail manure (QM) and the more commonly studied chicken manure, notably the typically higher crude protein content in QM due to the high-protein diets of quails. This provides a concrete reason for investigating QM as a distinct and potentially valuable substrate. Furthermore, we have added references to specific prior studies that investigated BSFL performance on various feeds, helping to create a better context for interpreting the performance variables (e.g., efficiency of conversion of ingested substrate) calculated in our study (line 62-68).

Line 76-79 is an example of, in my mind, insufficient introduction of current knowledge. It has now been described 'It is worth noting that substrates can influence the intestinal microbiota of BSF, these microbiotas can regulate the nutritional conversion efficiency of BSF by participating in the decomposition of complex organic matter and the synthesis of bioactive substances', but nothing about how the microbiota is influenced, which substrates, which microbes, or what it means to the BSF larvae.

Response: Agreed. We have thoroughly rewritten this section (line 69-83) to provide a more concrete and knowledge-grounded introduction. The new text specifies how different substrate types (e.g., high-fiber, high-protein) can select for distinct microbial communities. It also elaborates on the functional implications for the larvae, such as the role of these microbes in breaking down lignocellulose or influencing larval fat deposition, thereby directly linking substrate-induced microbial shifts to BSF growth and composition outcomes reported in existing studies.

Still, Line 42-69 is only marginally important to this study. Thus, this space could be used better.

Response: Thank you very much for your comments. We have condensed the general background information in the introductory section (line 42-56) and repurposed that valuable space to strengthen the parts of the Introduction that are directly relevant to our study.

As mentioned above, this is a highly detailed characterization of many aspects of BSF production on 3 selected feed substrates, and it may provide useful empirical information. Of course, it is up to the authors to decide how to frame their paper. I just hope my comments have been somewhat useful, despite being a bit critical. I cannot help recommending the authors take the above comments into consideration, thus I must recommend minor revisions once more.

Response: We are profoundly grateful for your critical and highly useful comments throughout the review process. The feedback has been instrumental in elevating the quality, clarity, and scholarly foundation of our work. We have taken the final set of comments fully into consideration. The revisions made in this round, particularly those aimed at strengthening the theoretical background and more clearly framing our study as a comprehensive empirical resource, directly address your suggestions. Thank you again for your valuable suggestions, as always.